# General and Efficient Steering of Unconditional Diffusion Models

Qingsong Wang [1]   Mikhail Belkin [1]   Yusu Wang [1]

## Abstract

Steering diffusion models toward conditions unseen during training typically requires either retraining with conditional inputs or per-step gradient computations, both of which incur substantial computational overhead. We present Noise-Aligned RFM Steering (NA-RFM), a general recipe for efficiently steering unconditional diffusion models without gradient guidance during inference, enabling fast controllable generation. The method combines two offline-computed signals: *noise alignment*, a high-noise correction from PCA statistics of the target examples and the full data, and *Recursive Feature Machine (RFM) activation steering*, which learns a target-discriminative direction from labeled forward-process activations. During sampling, noise alignment provides coarse control at high noise, while the RFM direction is reused over intermediate/late timesteps through lightweight activation edits. Experiments on CIFAR-10, ImageNet, CelebA, and fine-grained bird species show improved target accuracy over gradient-based post-hoc guidance baselines, improved FID on the class-guidance benchmarks, and substantial inference speedups. Code: https://github.com/isotrivial/na-rfm.

## 1. Introduction

Diffusion models (Ho et al., 2020; Song et al., 2021b) have become a dominant methodology for high-quality image synthesis. With classifier-free guidance (CFG) (Ho & Salimans, 2022), they can be guided effectively when the desired condition is built into training, as in class- or text-conditioned generation. Many applications instead require steering toward a condition that was unseen or unavailable

---

[1]Halıcıoğlu Data Science Institute, University of California San Diego, La Jolla, CA, USA. Correspondence to: Qingsong Wang <qswang92@gmail.com>.

*Proceedings of the 43rd International Conference on Machine Learning*, Seoul, South Korea. PMLR 306, 2026. Copyright 2026 by the author(s).

during diffusion-model training: a new object class, fine-grained species, or visual property specified only through examples. The key challenge is to steer a pretrained diffusion model toward such unseen conditions without retraining the model.

Popular approaches for steering unconditional models use classifier gradients to guide generation toward desired concepts. This includes both *training-based* noise-conditioned classifier guidance (Dhariwal & Nichol, 2021; Song et al., 2021b), which trains classifiers to predict labels from noisy images at different timesteps, and *training-free* gradient methods that use pre-existing off-the-shelf classifiers (He et al., 2024; Song et al., 2023; Chung et al., 2023; Bansal et al., 2023; Yu et al., 2023; Ye et al., 2024). Both noise-conditioned classifier guidance and training-free gradient guidance require per-step gradient computation, and many variants backpropagate through the diffusion model during inference.

In this paper, we develop Noise-Aligned RFM Steering (NA-RFM), a post-hoc method for steering pretrained diffusion models without inference-time gradients. Similar to CFG, NA-RFM is gradient-free during sampling, but its guidance signals are constructed after training the diffusion model. A lightweight offline stage uses target-vs-background examples to compute PCA statistics and learn activation-space steering directions via Recursive Feature Machines (RFMs) (Radhakrishnan et al., 2024). The online sampler then uses only diffusion-model forward passes, matrix-vector products, and activation-vector additions, yielding speedups over training-free gradient-based methods.

Two observations guide how we steer at different noise levels along the sampling trajectory:

(1) **Observation 1: high-noise coarse class structure.** At high noise, reverse trajectories already contain information predictive of the class ultimately generated. Gaussian/PCA analyses provide a tractable approximation to the diffusion model in this regime (Wang & Vastola, 2024; Li et al., 2024; 2026; Dodson et al., 2026). This motivates *noise alignment*: a high-noise pixel-space alignment signal computed from class-conditional and full-data PCA statistics.

(2) **Observation 2: transferable activation directions.** At

moderate and low noise levels, forward-process model activations provide a strong discriminative signal, and the corresponding activation direction remains aligned across forward-noised timesteps that match an intermediate/late reverse-sampling window. This motivates RFM activation steering: we learn a target-vs-rest Recursive Feature Machine (RFM) (Radhakrishnan et al., 2024) direction from forward-process activations, then *reuse* it over that sampling window.

The probing experiments in Section 3.1 motivate these observations. Existing high-noise Gaussian/PCA analyses support the first; for the second, we give a theoretical understanding of the stability of discriminative directions in the forward process through a simplified model.

Unconditional diffusion models provide a clean testbed for evaluation, as every target concept is unavailable as a training-time condition. On CIFAR-10 (Table 2), NA-RFM achieves 96.6% guidance accuracy compared to 77.1% for the state-of-the-art training-free gradient baseline TFG (Ye et al., 2024), and also outperforms noise-conditioned classifier guidance (86.0%), while delivering strong image quality (FID 41.4 vs. 73.9 vs. 41.9) and a $16\times$ inference speedup over TFG. The accuracy gains persist on ImageNet $256 \times 256$, multi-attribute CelebA guidance, and out-of-distribution fine-grained bird species (Section 5). Further experiments show NA-RFM extends to transformer-based latent diffusion (SiT-XL/2 (Ma et al., 2024); Appendix G.1) and to Stable Diffusion 1.5 (Rombach et al., 2022) for depth-of-field control, a photographic property that can be difficult to specify reliably with text prompts alone (Fortes et al., 2025) (Appendix G.2).

In summary, NA-RFM combines high-noise noise alignment with intermediate/late RFM activation steering to provide post-hoc steering of pretrained diffusion models with gradient-free online inference.

## 2. Background on Diffusion Models and Guidance

Diffusion models (Ho et al., 2020; Song et al., 2021b) are trained by learning to denoise noisy inputs at various timesteps; generation then samples through the learned denoising process. The **forward (noising) process** linearly mixes clean data $x_0$ with Gaussian noise over timesteps $t = 0, \dots, T$:

$$x_t = \alpha_t x_0 + \beta_t \epsilon, \quad \epsilon \sim \mathcal{N}(0, I), \tag{1}$$

where $\alpha_t$ and $\beta_t$ are the signal and noise coefficients, with $\alpha_0 = 1$, $\beta_0 = 0$ (clean data) and $\alpha_T \approx 0$, $\beta_T \approx 1$ (pure noise). We define the *noise-to-signal ratio*

$$\sigma_t := \frac{\beta_t}{\alpha_t}, \qquad \tilde{x}_t := \frac{x_t}{\alpha_t} = x_0 + \sigma_t \epsilon. \tag{2}$$

Equivalently, after dividing by the signal coefficient $\alpha_t$, $\sigma_t$ is the standard deviation of the additive noise in $\tilde{x}_t$. We use this $\sigma_t$ as the reporting convention for guidance windows. Most main U-Net experiments, including CIFAR-10 and ADM ImageNet/Birds, use a VP/DDPM parameterization with $T = 1000$ training timesteps and a linear variance schedule; detailed conversions are given in Appendix H.1.

A key quantity for guidance methods is the **denoised estimate**, i.e., the denoiser output after converting the predicted noise to an estimated clean image:

$$\hat{x}_0^{(t)} = \frac{x_t - \beta_t \, \epsilon_\theta(x_t, t)}{\alpha_t}, \tag{3}$$

This maps the current noisy state $x_t$ to the model's prediction of the clean image $x_0$, and therefore gives a direct estimate of the final output at any intermediate timestep. For the main experiments in this paper, we sample with DDIM (Song et al., 2021a). The DDIM update with stochasticity level $\eta$ is:

$$x_{t-1} = \alpha_{t-1} \hat{x}_0^{(t)} + \sqrt{\beta_{t-1}^2 - \gamma_t^2} \, \epsilon_\theta(x_t, t) + \gamma_t z, \tag{4}$$

where $\gamma_t = \eta \cdot \frac{\beta_{t-1}}{\beta_t} \sqrt{1 - \frac{\alpha_t^2}{\alpha_{t-1}^2}}$, and $z \sim \mathcal{N}(0, I)$. Setting $\eta = 0$ recovers the deterministic ODE sampler.

**Remark.** For a fixed noise schedule, noise prediction, denoised estimate prediction, velocity prediction, and score prediction are related by simple transformations. We write the main equations in DDPM/DDIM notation because this is the parameterization used by our main U-Net experiments.

**Classifier Guidance** (Dhariwal & Nichol, 2021) steers generation by modifying the noise prediction using gradients from a noise-conditioned classifier $p_\phi(y|x_t)$ trained on noisy images at all timesteps:

$$\tilde{\epsilon}_\theta(x_t, t, y) = \epsilon_\theta(x_t, t) - \beta_t \cdot w \nabla_{x_t} \log p_\phi(y|x_t). \tag{5}$$

The gradient $\nabla_{x_t} \log p_\phi(y|x_t)$ gives a direction in pixel space that increases the probability of class $y$, applied to the noise prediction during sampling. This approach requires training noise-conditioned classifiers for all timesteps and backpropagating through the classifier at *every* denoising step during inference.

**Classifier-Free Guidance (CFG)** (Ho & Salimans, 2022) eliminates the need for an auxiliary classifier by training a conditional model $\epsilon_\theta(x_t, t, c)$ with random condition dropout. At inference, guidance is achieved by modifying the noise prediction through interpolation:

$$\tilde{\epsilon}_\theta = \epsilon_\theta(x_t, t, \varnothing) + w \cdot (\epsilon_\theta(x_t, t, c) - \epsilon_\theta(x_t, t, \varnothing)). \tag{6}$$

While elegant and widely used, this requires conditioning at training time, providing limited post-hoc controllability for attributes not seen during training.

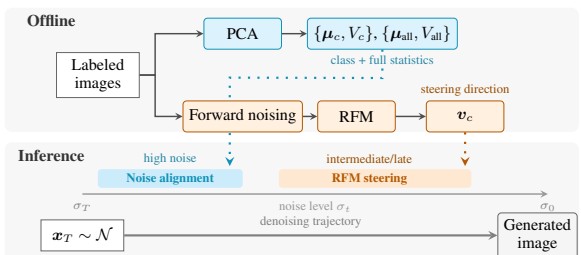

*Figure 1.* **Overview of Noise-Aligned RFM Steering.** Offline, we compute class-conditional PCA statistics (cyan) and RFM directions from forward-process activations (orange). At inference, noise alignment supplies the high-noise correction and RFM steering supplies intermediate/late activation-space control, with no classifier gradients.

**Training-Free Gradient-based Guidance.** To enable post-hoc control without retraining, several methods use off-the-shelf classifiers trained on clean images and backpropagate guidance from either $x_t$ or the denoised estimate $\hat{x}_0^{(t)}$ at each step (Bansal et al., 2023; Yu et al., 2023; Ye et al., 2024). Representative lines include inverse-problem solvers such as DPS, LGD (Chung et al., 2023; Song et al., 2023), iterative refinement strategies like FreeDoM, and variants that guide through $\hat{x}_0^{(t)}$ or add backward optimization (e.g., MPGD, UGD) (Yu et al., 2023; He et al., 2024; Bansal et al., 2023; Ye et al., 2024). These methods often face weak or misaligned classifier gradients, especially at high noise, sometimes addressed with extra refinement steps; their per-step backpropagation also makes sampling substantially slower than unconditional generation.

## 3. Method: Noise-Aligned RFM Steering

NA-RFM separates *offline construction* from *gradient-free online sampling*. *Offline*, for each target concept $c$ such as a class or example-defined visual property, we use examples to compute the PCA statistics for high-noise guidance and learn an RFM direction $v_c$ from low-noise forward-process activations in a selected network block. *During sampling*, the PCA statistics give a pixel-space correction at high noise, and the RFM direction is applied to the selected block over the intermediate/late sampling window. Figure 1 summarizes the pipeline. We next motivate the design choices and give the implementation details.

### 3.1. Empirical Observations

We use the pretrained CIFAR-10 diffusion U-Net as a controlled setting for identifying which U-Net activations carry class-discriminative information at which noise levels. For a fixed U-Net block and noise level, we train 10-way linear probes on 10,000 activation examples and report accuracy on a held-out 20% split, comparing two activation collections with different label sources:

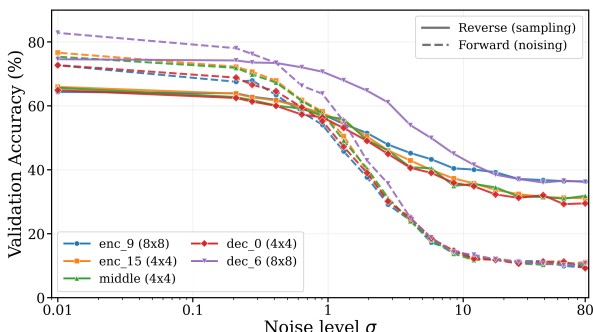

*Figure 2.* **Linear probe accuracy on forward vs. reverse diffusion activations.** Solid curves use activations recorded along unconditional reverse sampling trajectories and labeled by the final generated class assigned by the external evaluation classifier. Dashed curves use activations from labeled training images corrupted to the corresponding noise level and labeled by the image label. We report five representative U-Net blocks.

(a) **Reverse-trajectory activation probe.** We run the diffusion model, record U-Net activations along reverse ODE sampling trajectories, and label each recorded activation by the *final generated class* assigned to its terminal image by the external evaluation classifier. This probe asks whether activations at a noisy reverse step already predict the final generated class.

(b) **Forward-process activation probe.** We take labeled CIFAR-10 images, corrupt each image to the same noise levels using Equation (1), record activations from the same U-Net blocks, and use the clean-image class as the probe label. This probe asks at which noise levels labeled examples give separable activation features for offline direction learning.

These probes separate two possible sources for learning an activation-space steering direction $v_c$ with a lightweight RFM. Reverse trajectories show activations visited by the sampler, but using them for direction learning would require generating and labeling trajectories for each target concept, which is costly when the target is rare or hard to generate. Forward-process activations are cheaper: one corruption and one denoiser pass per labeled image. The probes below test where these examples provide separable activation features, which determines where we learn $v_c$. We report representative deeper blocks, including the $4 \times 4$ and $8 \times 8$ blocks where the class signal is strongest.

**Observation 1: high-noise coarse class structure.** The reverse-trajectory probe in Figure 2 shows that activations remain class-informative even at high noise. Thus, high-noise reverse activations already contain coarse information about the final generated class. At the same high noise levels, forward-process activations are near chance for class probing, so we do not learn RFM directions from them. For high noise, we therefore impose guidance by mimicking

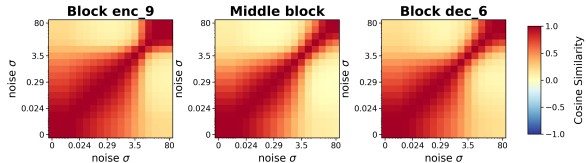

*Figure 3.* **Temporal transfer of activation directions.** Each entry reports the cosine similarity between activation directions collected from the forward process at two noise levels. Both axes are ordered by noise level $\sigma_t$, from low noise to high noise. All entries are non-negative, and the low noise levels activations are more aligned with intermediate noise levels than with high noise levels across the blocks.

CFG with a Gaussian/PCA approximation of the diffusion model: class-conditional and full-data PCA denoisers provide the pixel-space correction used by noise alignment in Section 3.2.

**Observation 2: transferable activation directions.** As the noise level decreases, forward-process activations become strongly class-discriminative, exceeding 80% probe accuracy near the end of the trajectory in Figure 2. These moderate/low-noise forward activations are the labeled activation source used later for RFM direction learning. The practical question is whether a direction learned once, at a reference noise level, can be reused over later sampling noise levels. We test this by comparing the cosine similarity of activation directions collected from the forward process at different noise levels. Figure 3 shows that these activation directions remain aligned across the intermediate/late window across blocks while the low noise levels are poorly aligned with high noise level, matching the weak high-noise forward probes in Figure 2. This motivates learning an RFM steering direction offline from low-noise forward activations and reuse it over the intermediate noise window. Proposition A.1 gives a simple model in which a class-discriminative direction is robust to noising. The sampler below therefore uses noise alignment at high noise and RFM activation steering over the intermediate/late range.

### 3.2. Noise Alignment in the High-Noise Window

Noise alignment is the high-noise update suggested by Observation 1. Observation 1 shows that coarse class information already appears at high noise.

Although reverse-trajectory activations are informative in this regime, using them for target-specific direction learning would require generating trajectories for each target concept. This is especially costly when the target is rare or hard to sample from the unguided model. In the same high-noise regime, linear-Gaussian/PCA approximations give a tractable model of the denoising map: fine details are suppressed, while class-level means and leading principal

directions can still supply a coarse steering signal (Wang & Vastola, 2024; Li et al., 2024; 2026; Dodson et al., 2026). We therefore use the difference between a class-conditional PCA denoiser and an unconditional PCA denoiser as a pixel-space guidance signal. Let $\tilde{\boldsymbol{x}}_t = \boldsymbol{x}_t/\alpha_t = \boldsymbol{x}_0 + \sigma_t \boldsymbol{\epsilon}$, where $\sigma_t = \beta_t/\alpha_t$ is the noise-to-signal ratio from Equation (2). Following Li et al. (2024; 2026), the PCA denoiser of the subclass data at $\tilde{\boldsymbol{x}}_t$ is given by:

$$D_c(\tilde{\boldsymbol{x}}_t; \sigma_t) = \boldsymbol{\mu}_c + V_c \text{diag}\left(\frac{\nu_{cj}}{\nu_{cj} + \sigma_t^2}\right) V_c^\top (\tilde{\boldsymbol{x}}_t - \boldsymbol{\mu}_c), \quad (7)$$

where $\boldsymbol{\mu}_c$ is the class mean, columns of $V_c$ are the retained principal directions, and $\{\nu_{cj}\}$ are the corresponding covariance eigenvalues. Similarly, we compute an unconditional denoiser $D_{\text{all}}(\tilde{\boldsymbol{x}}_t; \sigma_t)$ using full-dataset statistics $(\boldsymbol{\mu}_{\text{all}}, V_{\text{all}}, \{\nu_{\text{all},j}\})$.

The guidance signal is the difference between conditional and unconditional denoisers:

$$\boldsymbol{g}_t^{\text{PCA}} = D_c(\tilde{\boldsymbol{x}}_t; \sigma_t) - D_{\text{all}}(\tilde{\boldsymbol{x}}_t; \sigma_t). \quad (8)$$

The update uses only precomputed class and full-data statistics, and targets the high-noise regime where the forward activations fail to provide useful signals.

### 3.3. RFM Direction Discovery and Activation Steering

RFM steering provides the second, intermediate/late, guidance mechanism in NA-RFM. Guided by Observation 2, we use the temporal stability of forward-process activation directions to learn a target direction offline and reuse it during sampling, rather than learning a separate direction at each timestep. This choice also avoids target-specific reverse trajectories: moderate/low-noise forward activations are already class-discriminative (Figure 2), and the activation direction remains aligned over the intermediate/late sampling window (Figure 3). We use Recursive Feature Machines (RFMs) (Radhakrishnan et al., 2024) for this direction-learning step because they learn task-adapted feature metrics from limited examples in high-dimensional activation spaces, while the diffusion model itself remains fixed.

**Offline Activation Collection.** Fix a low-noise reference timestep $t_{\text{R}}$ and a U-Net block $\ell$. We typically use the last encoder block before the bottleneck; block-selection ablations are reported in Appendix C. Given labeled images $\{(\boldsymbol{x}_i, y_i)\}_{i=1}^N$, draw $\boldsymbol{\epsilon}_i \sim \mathcal{N}(\boldsymbol{0}, \boldsymbol{I})$ and collect activations by forward noising each image to $t_{\text{R}}$:

$$\begin{aligned} \boldsymbol{x}_{t_{\text{R}}}^{(i)} &= \alpha_{t_{\text{R}}} \boldsymbol{x}_i + \beta_{t_{\text{R}}} \boldsymbol{\epsilon}_i, \\ \boldsymbol{h}_i &= \text{vec}\left(\phi_\ell(\boldsymbol{x}_{t_{\text{R}}}^{(i)}, t_{\text{R}})\right) \in \mathbb{R}^{d_h}, \end{aligned} \quad (9)$$

where $\phi_\ell$ extracts the block-$\ell$ activation tensor $\boldsymbol{H}_i^{(\ell)}$, and $\boldsymbol{h}_i$ is its flattened form in dimension $d_h = C_\ell H_\ell W_\ell$. The offline stage outputs one unit direction $\boldsymbol{v}_c$ for each target class; when used inside the U-Net, $\boldsymbol{v}_c$ is reshaped back to the tensor shape of block $\ell$. For target class $c$, we use binary labels $z_i^c = +1$ if $y_i = c$ and $z_i^c = -1$ otherwise.

**RFM Training.** Given flattened activations $\{\boldsymbol{h}_i\}$ and binary labels $\{z_i^c\}$, we train one RFM model for class $c$ versus the remaining classes or background data. Following the RFM feature-learning mechanism of Radhakrishnan et al. (2024), the RFM model maintains a Mahalanobis feature metric $M$ over activation space; directions with larger weight under $M$ are treated as more important features. We update this metric iteratively. Starting from $M^{(0)} = I$, iteration $r$ builds a Laplacian kernel

$$K_{ij}^{(r)} = \exp(-\gamma\, d_{M^{(r)}}(\boldsymbol{h}_i, \boldsymbol{h}_j)),$$
$$d_{M^{(r)}}(\boldsymbol{h}_i, \boldsymbol{h}_j) = \|(M^{(r)})^{1/2}(\boldsymbol{h}_i - \boldsymbol{h}_j)\|_2,$$

solves kernel ridge regression to obtain a predictor $f^{(r)}$, and sets the next metric to its average gradient outer product (AGOP):

$$M^{(r+1)} = M_{\text{AGOP}}^{(r)} = \frac{1}{N} \sum_i \nabla_{\boldsymbol{h}} f^{(r)}(\boldsymbol{h}_i) \nabla_{\boldsymbol{h}} f^{(r)}(\boldsymbol{h}_i)^\top.$$

Thus each predictor defines the metric used by the next predictor, progressively emphasizing activation directions that separate the target from the rest. In our setting, directly forming this matrix can be prohibitively expensive because $d_h = C_\ell H_\ell W_\ell$ is large while $N \ll d_h$. At iteration $r$, we stack the activation gradients as rows of $G^{(r)} \in \mathbb{R}^{N \times d_h}$ and compute the smaller $N \times N$ matrix $G^{(r)} G^{(r)\top}/N$, using the same sample-space principle as eigenfaces/PCA (Turk & Pentland, 1991). Its nonzero eigenspace determines the corresponding nonzero eigenspace of $G^{(r)\top} G^{(r)}/N$, which is the part of the AGOP used for the next RFM metric update and for the final steering direction.

**Forming the Steering Direction.** After the validation-selected RFM iteration, stack the activation gradients as rows of $G \in \mathbb{R}^{N \times d_h}$ and let $A = G^\top G/N$ be the final activation-space AGOP. We obtain the leading eigenpairs $(\rho_j, \boldsymbol{u}_j)$ of $A$ by the sample-space computation above; Appendix B gives the algebra. The sign of each eigenvector is arbitrary, so following Beaglehole et al. (2026), we compute the Pearson correlation between each eigenvector's projection scores and the target-vs-rest labels. Let $s_{c,j} \in \{\pm 1\}$ be the sign that makes this correlation nonnegative for eigenvector $j$. The class direction is the eigenvalue-weighted, sign-corrected top-$k$ combination:

$$\tilde{\boldsymbol{v}}_c = \sum_{j=1}^{k} \frac{\rho_j}{\sum_{r=1}^{k} \rho_r} s_{c,j} \boldsymbol{u}_j, \qquad \boldsymbol{v}_c = \tilde{\boldsymbol{v}}_c / \|\tilde{\boldsymbol{v}}_c\|_2,$$

with $k \in \{1, 3, 5\}$. This unit vector is the target-specific steering direction used in the activation update below.

**Online Activation Steering.** During sampling, the learned direction is fixed; the only online operation is to reshape it to the selected block and add it to that block's activation. Let $\boldsymbol{V}_c^{(\ell)}$ denote $\boldsymbol{v}_c$ reshaped to the tensor shape of block $\ell$. When RFM steering is active, we run a denoiser pass in which the activation tensor at block $\ell$ is replaced by

$$\boldsymbol{H}_{\text{steered}}^{(\ell)} = \boldsymbol{H}^{(\ell)} + w_{\text{RFM}} \|\boldsymbol{H}^{(\ell)}\|_F \boldsymbol{V}_c^{(\ell)}, \qquad (10)$$

where $w_{\text{RFM}}$ is the steering strength and $\|\boldsymbol{V}_c^{(\ell)}\|_F = \|\boldsymbol{v}_c\|_2 = 1$. Here $\|\cdot\|_F$ denotes the Frobenius norm. The activation-norm factor keeps the guidance magnitude proportional to the current activation scale. An amplification step analogous to CFG further improves RFM steering. Specifically, the steered pass produces a noise prediction $\hat{\epsilon}_{\text{rfm}}$ and denoised estimate $\hat{\boldsymbol{x}}_{0,\text{rfm}}$. Let $\hat{\boldsymbol{x}}_{0,\text{base}}$ be the unsteered denoised estimate from the same sampling step. We express the activation edit in clean-image coordinates by extrapolating from the base estimate toward the steered one:

$$\hat{\boldsymbol{x}}_0 \leftarrow \hat{\boldsymbol{x}}_{0,\text{base}} + s \cdot (\hat{\boldsymbol{x}}_{0,\text{rfm}} - \hat{\boldsymbol{x}}_{0,\text{base}}), \qquad (11)$$

where $s$ is the amplification scale. When $s = 1$, the sampler uses the steered denoiser output directly; larger values strengthen the activation-space edit.

### 3.4. Inference

At inference time, NA-RFM imposes guidance without gradient computation. Algorithm 1 summarizes the resulting sampler. Each sample step first runs the denoiser once to obtain an unsteered estimate, then applies the guidance mechanisms whose noise windows are active at the current noise level $\sigma_t$. When noise alignment is active, the step adds the precomputed PCA correction $D_c(\tilde{\boldsymbol{x}}_t; \sigma_t) - D_{\text{all}}(\tilde{\boldsymbol{x}}_t; \sigma_t)$ directly to $\hat{\boldsymbol{x}}_0$. When RFM steering is active, the step uses one additional denoiser evaluation with the block-$\ell$ activation replaced by $\boldsymbol{H}_{\text{steered}}^{(\ell)}$, then applies the affine update in Equation (11). No step backpropagates through a classifier or through the diffusion model.

**Guidance Windows.** We schedule both mechanisms using the noise parameter $\sigma_t$ from Equation (2). The *noise-alignment (NA) window* is active when $\sigma_t \geq \sigma_{\text{end}}$, the initial high-noise part of the trajectory where we apply the PCA-based coarse correction. The *RFM window* is active when $\sigma_t \in [\sigma_R^{\text{lo}}, \sigma_R^{\text{hi}}]$, the intermediate/low-noise range where the activation direction shows stable alignment. The dataset-specific ranges are reported in Appendix H.

**Computational complexity.** Offline preparation uses target-specific labeled examples but does not retrain the diffusion

**Algorithm 1** Guided sampling with noise alignment and RFM steering. For strided DDIM sampling, $t-1$ denotes the next lower-noise scheduler index.

**Require:** PCA denoisers $D_c, D_{\text{all}}$; RFM tensor direction $\boldsymbol{V}_c^{(\ell)}$ for block $\ell$

**Require:** Noise alignment coefficient $\lambda$; Noise alignment window $\sigma_t \geq \sigma_{\text{end}}$

**Require:** RFM steering coefficient $w_{\text{RFM}}$; RFM steering window $[\sigma_{\text{R}}^{\text{lo}}, \sigma_{\text{R}}^{\text{hi}}]$; RFM amplification scale $s$

1: $\boldsymbol{x}_T \sim \mathcal{N}(0, \boldsymbol{I})$
2: **for** $t = T, \ldots, 1$ **do**
3:     $(\alpha_t, \beta_t, \sigma_t) \leftarrow$ scheduler coefficients with $\sigma_t = \beta_t / \alpha_t$
4:     Run $\boldsymbol{\epsilon}_\theta(\boldsymbol{x}_t, t)$ once to obtain $\hat{\boldsymbol{\epsilon}}$ and block-$\ell$ activation $\boldsymbol{H}^{(\ell)}$
5:     $\hat{\boldsymbol{x}}_{0,\text{base}} \leftarrow (\boldsymbol{x}_t - \beta_t \hat{\boldsymbol{\epsilon}}) / \alpha_t$
6:     $\hat{\boldsymbol{x}}_0 \leftarrow \hat{\boldsymbol{x}}_{0,\text{base}}$
7:     ▷ RFM steering with amplification
8:     **if** $\sigma_t \in [\sigma_{\text{R}}^{\text{lo}}, \sigma_{\text{R}}^{\text{hi}}]$ **then**
9:         $\boldsymbol{H}_{\text{steered}}^{(\ell)} \leftarrow \boldsymbol{H}^{(\ell)} + w_{\text{RFM}} \|\boldsymbol{H}^{(\ell)}\|_F \boldsymbol{V}_c^{(\ell)}$
10:        Run $\boldsymbol{\epsilon}_\theta(\boldsymbol{x}_t, t)$ with the block-$\ell$ activation replaced by $\boldsymbol{H}_{\text{steered}}^{(\ell)}$ to obtain $\hat{\boldsymbol{\epsilon}}_{\text{rfm}}$
11:        $\hat{\boldsymbol{x}}_{0,\text{rfm}} \leftarrow (\boldsymbol{x}_t - \beta_t \hat{\boldsymbol{\epsilon}}_{\text{rfm}}) / \alpha_t$
12:        $\hat{\boldsymbol{x}}_0 \leftarrow \hat{\boldsymbol{x}}_{0,\text{base}} + s \cdot (\hat{\boldsymbol{x}}_{0,\text{rfm}} - \hat{\boldsymbol{x}}_{0,\text{base}})$
13:     **end if**
14:     ▷ Noise Alignment
15:     **if** $\sigma_t \geq \sigma_{\text{end}}$ **then**
16:        $\tilde{\boldsymbol{x}}_t \leftarrow \boldsymbol{x}_t / \alpha_t$
17:        $\boldsymbol{g}_t^{\text{PCA}} \leftarrow D_c(\tilde{\boldsymbol{x}}_t; \sigma_t) - D_{\text{all}}(\tilde{\boldsymbol{x}}_t; \sigma_t)$
18:        $\hat{\boldsymbol{x}}_0 \leftarrow \hat{\boldsymbol{x}}_0 + \lambda \boldsymbol{g}_t^{\text{PCA}}$
19:     **end if**
20:     $\hat{\boldsymbol{\epsilon}} \leftarrow (\boldsymbol{x}_t - \alpha_t \hat{\boldsymbol{x}}_0) / \beta_t$
21:     $\boldsymbol{x}_{t-1} \leftarrow \alpha_{t-1} \hat{\boldsymbol{x}}_0 + \beta_{t-1} \hat{\boldsymbol{\epsilon}}$ {DDIM, $\eta=0$}
22: **end for**
23: **return** $\boldsymbol{x}_0$

model. For a set of target classes, the offline work is to compute class and unconditional PCA statistics, collect one shared forward-process activation set, and train one RFM direction per target class. For high-dimensional, low-sample data, both linear-algebra steps use compact sample-space computations: PCA uses the sample matrix without forming the image-space covariance, and the RFM AGOP eigenspace is recovered from $GG^\top / N$ rather than $G^\top G / N$. Online, a baseline step uses one denoiser pass; a noise-alignment step adds PCA matrix-vector products; and an RFM-active step adds one steered denoiser pass. No online step backpropagates through a classifier or through the diffusion model. Table 3 reports measured offline preparation time and online sampling cost for the ImageNet setting.

## 4. Related Work

**Activation Steering in Diffusion Models.** In diffusion models, the U-Net bottleneck features ("h-space") have been shown to act as a semantic latent space and support linear, interpretable edits (Kwon et al., 2023); follow-up work uses h-space feature manipulation for training-free content injection and editing (Jeong et al., 2024). Those methods are mainly developed for input-specific editing, often using DDIM inversion to obtain the latent for a given image. Other text-to-image editing methods manipulate cross-attention maps (Hertz et al., 2023; Patashnik et al., 2023) or invert prompts back into the conditioning space (Mahajan et al., 2024). NA-RFM instead learns a class-level direction offline from labeled examples, applies the same direction to each generation trajectory, and avoids inference-time backpropagation through the diffusion network.

**Semantic Structure and Early Concept Emergence in Diffusion.** Prior work has shown that semantic structure appears early along the diffusion trajectory. Hertz et al. (2023) show that cross-attention maps encode scene layout during the first few denoising steps; Patashnik et al. (2023) use this structure to localize shape edits; Tinaz et al. (2026) track the emergence and evolution of interpretable concepts along the reverse trajectory; and Li et al. (2025); Wang et al. (2026) show that the initial noise seed carries high-level compositional cues. Closely related to our method, Meng et al. (2024) show that intermediate blocks of diffusion U-Nets, evaluated on noisy inputs, can serve as *discriminative feature* extractors for dense prediction tasks, with quality varying across blocks and timesteps. This literature supports probing U-Net activations as semantic features at noisy timesteps in Figure 2. Figure 3 tests the additional temporal-transfer property used by NA-RFM: forward-process activation directions remain aligned across the intermediate/late sampling range.

**Steering Vectors in Language Models.** Related work in large language models steers pretrained generators by adding learned directions to internal activations (Beaglehole et al., 2026; Turner et al., 2023; Zou et al., 2023). These directions, extracted from contrasting prompts, linear probes, or RFM, are added at one or more transformer layers to bias generation without gradient computation. NA-RFM adapts this idea to diffusion models, where activations are high-dimensional feature maps and the same block is evaluated across many noise levels. These differences motivate the sample-space RFM computation and the temporal-transfer analysis in Figure 3.

## 5. Experiments

We evaluate NA-RFM on four post-hoc steering settings used by TFG (Ye et al., 2024): CIFAR-10 label guidance,

*Table 1.* **Overall comparison with gradient-based post-hoc baselines.** We compare Noise-Aligned RFM Steering against TFG (Ye et al., 2024) and reported baselines across the four benchmark tasks. Each cell reports *accuracy (%) / quality metric*; the quality metric is FID↓ except for CelebA-HQ, where we report log-KID↓. TFG results are from Ye et al. (2024). TFG-4 is not reported for CelebA-HQ; Table 4 gives stratified TFG accuracies. Bold and underline mark the best and second-best target accuracy, respectively; quality metrics are reported alongside accuracy.

| Task | DPS | LGD | FreeDoM | MPGD | UGD | TFG-1 | TFG-4 | NA-RFM |
|---|---|---|---|---|---|---|---|---|
| CIFAR-10 ($\uparrow$, $\downarrow$) | 50.1 / 172 | 32.2 / 102 | 34.8 / 135 | 38.0 / 88 | 45.9 / 94 | 52.0 / 92 | 77.1 / 73.9 | **96.6** / 41.4 |
| ImageNet ($\uparrow$, $\downarrow$) | 38.8 / 193 | 11.5 / 210 | 19.7 / 200 | 6.8 / 239 | 25.5 / 205 | 40.9 / 176 | 59.8 / 165 | **75.8** / 98 |
| Gender+Age ($\uparrow$, $\downarrow$) | 71.6 / -4.3 | 52.0 / -5.1 | 68.7 / -3.9 | 68.6 / -4.8 | 75.1 / -4.4 | 75.2 / -3.9 | – | **96.0** / -2.0 |
| Gender+Hair ($\uparrow$, $\downarrow$) | 73.0 / -3.9 | 55.0 / -5.0 | 67.1 / -3.5 | 63.9 / -4.3 | 71.3 / -4.1 | 76.0 / -3.6 | – | **83.3** / -2.4 |
| Fine-grained ($\uparrow$, $\downarrow$) | 0.0 / 348 | 0.5 / 246 | 0.6 / 258 | 0.6 / 249 | 1.1 / 255 | 1.3 / 256 | 2.2 / 259 | **14.1** / 72 |

ImageNet $256 \times 256$ label guidance, CelebA-HQ multi-attribute guidance, and fine-grained bird-species guidance using an ImageNet backbone. In each setting, the target condition is unseen or unavailable as a training-time condition for the diffusion model, so we compare against post-hoc guidance baselines. Our primary baseline is TFG, a recent gradient-based post-hoc guidance method; where available, we report both TFG-1 and TFG-4, corresponding to $N_{recur} = 1$ and $N_{recur} = 4$. Comparing with the training-free methods, our NA-RFM uses target-vs-background examples to construct guidance signals offline and then samples without guidance-classifier evaluations or classifier-gradient backpropagation. Table 1 gives the cross-task summary; the following subsections report dataset-specific results, with qualitative grids and related diagnostics collected in Appendix E.

Across the four main settings, NA-RFM provides stronger target control than TFG-4. The gains are largest on CIFAR-10, ImageNet, and fine-grained species guidance, where target accuracy increases from 77.1% to 96.6%, from 59.8% to 75.8%, and from 2.2% to 14.1%, respectively, with improved FID in all three cases. Overall, using example data and lightweight offline preparation gives stronger target control than guidance from off-the-shelf classifier gradients while avoiding online gradient computation.

**Experimental protocol.** We use CIFAR-10 (Krizhevsky et al., 2009), ImageNet (Russakovsky et al., 2015) classes 111, 222, 333, and 444 following Ye et al. (2024), CelebA-HQ (Karras et al., 2018), and Birds-525 for fine-grained species guidance. For CIFAR-10, we use the improved DDPM U-Net (Nichol & Dhariwal, 2021); for ImageNet and Birds-525, the unconditional ADM model (Dhariwal & Nichol, 2021); and for CelebA-HQ, a DDPM trained on CelebA-HQ. All our main U-Net experiments use deterministic DDIM (Song et al., 2021a) sampling with 100 steps. Implementation details, including checkpoints, activation blocks, guidance strengths, RFM fitting parameters, and per-dataset settings, are in Appendix H.

**Labels, evaluators, and metrics.** For CIFAR-10, ImageNet, and Birds-525, the offline guidance construction

*Table 2.* **CIFAR-10 label guidance comparison.** NA-RFM achieves the highest accuracy and the best FID among external guidance baselines. Timing measured on A100 GPU for 16 samples with 100 DDIM steps.

| Method | Acc. $\uparrow$ | FID $\downarrow$ | Time |
|---|---|---|---|
| TFG-4 | 77.1% | 73.9 | 101.7s |
| Classifier-Guidance | 86.0% | 41.9 | 6.9s |
| Noise align. only ($\lambda$=3) | 62% | 99 | **5.8s** |
| Noise align. only ($\lambda$=8) | 80% | 120 | **5.8s** |
| RFM-only (all steps) | 94.8% | **40.3** | 6.8s |
| **NA-RFM** | **96.6%** | 41.4 | 6.2s |

uses the available dataset labels. For CelebA-HQ, we follow the TFG protocol: attribute labels are assigned using the classifier used for TFG guidance, while generated images are evaluated with the provided evaluation classifier. In all settings, reported accuracy is measured by the benchmark evaluation classifier, not by an online objective optimized by NA-RFM. We report target accuracy, FID (Heusel et al., 2017) for image quality, and log-KID (Bińkowski et al., 2018) for CelebA-HQ. Additional diversity and evaluator-robustness audits are reported in Appendix F.

**5.1. CIFAR-10: Controlled Benchmark**

Table 2 reports CIFAR-10 label-guidance results. NA-RFM reaches **96.6%** target accuracy with FID 41.4, compared with 77.1% accuracy and FID 73.9 for TFG-4, and 86.0% accuracy and FID 41.9 for classifier guidance (Nichol & Dhariwal, 2021). It is also **16× faster** than TFG in the reported setting. The component rows show the roles of the two mechanisms: RFM-only guidance already provides most of the accuracy and quality gain, while adding high-noise noise alignment improves accuracy from 94.8% to 96.6% with a small FID change from 40.3 to 41.4.

Notably, noise alignment alone is competitive with TFG-1 in accuracy, while being much faster and simpler to implement. Figure 4 shows the broader noise-alignment-only sweep: stronger alignment improves target control but degrades FID. In the final sampler, this high-noise correction pro-

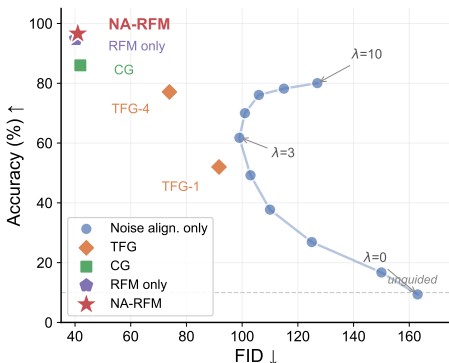

*Figure 4.* **Accuracy–FID trade-off for noise alignment only.** Points show noise-alignment-only results at varying $\lambda$; the dashed line indicates the Pareto frontier. Stronger noise alignment improves accuracy but degrades FID. The full NA-RFM result improves over this noise-alignment-only trade-off by adding RFM activation steering.

*Table 3.* **ImageNet offline and online computation.** A100 wall-clock timing for the reported four-class ImageNet evaluation (1024 images), excluding model/classifier pretraining. Offline columns are measured in minutes, the online column reports measured seconds per generated image at batch size 4, and the total column reports hours for the full 1024-image evaluation. The NA-RFM offline setup is split into shared activation collection and measured PCA+RFM computation.

| Method | Act. collect. (min.) | PCA+RFM (min.) | Online (sec./img) | Total (h) |
|---|---|---|---|---|
| TFG-4 | 0 | 0 | 79.70 | 22.7 |
| **NA-RFM** | 15.42 | 10.03 | **7.90** | **2.67** |

vides coarse control, while intermediate/late RFM steering supplies the main accuracy and quality gains. Appendix E.1 gives per-class breakdowns.

### 5.2. ImageNet: Scaling to Higher Resolution

ImageNet tests whether the same post-hoc guidance signals scale to a high-resolution unconditional ADM model (Dhariwal & Nichol, 2021). Following TFG (Ye et al., 2024), we evaluate classes 111, 222, 333, and 444 with 256 samples per class. As shown in Table 1, NA-RFM achieves **75.8%** average target accuracy, compared with 59.8% for TFG-4.

Table 3 separates one-time preparation from per-image sampling for the full four-class ImageNet evaluation ($4 \times 256 = 1024$ images), excluding model and classifier pretraining. NA-RFM spends 25.45 minutes on activation collection and PCA/RFM computation, then samples at 7.90 seconds per image at batch size 4, compared with 79.70 seconds per image for TFG-4 under the same measurement.

Equivalently, for $n$ generated images, the measured total time in seconds is $T_{\text{NA-RFM}}(n) = 1527.16 + 7.90n$ and $T_{\text{TFG-4}}(n) = 79.70n$. The offline setup is amortized after

*Table 4.* **CelebA multi-attribute guidance.** Comparison with the per-combination TFG accuracies reported by Ye et al. (2024) for gender+hair and gender+age guidance (256 samples). NA-RFM is higher on 7 of 8 combinations, including two 100% accuracy cases.

| Attributes | TFG | NA-RFM |
|---|---|---|
| *Gender + Hair* | | |
| Female + Non-Blond | **92.2%** | 86.4% |
| Female + Blond | 72.7% | **80.1%** |
| Male + Non-Blond | 89.8% | **98.4%** |
| Male + Blond | 46.7% | **68.4%** |
| Average | 75.4% | **83.3%** |
| *Gender + Age* | | |
| Young + Female | 92.9% | **100.0%** |
| Old + Female | 73.6% | **85.2%** |
| Young + Male | 93.6% | **98.8%** |
| Old + Male | 69.1% | **100.0%** |
| Average | 82.3% | **96.0%** |

roughly 22 generated images; on the 1024-image evaluation, total time drops from 22.7 to 2.67 hours.

**Per-Class Analysis.** Accuracy is high on *nematode* (83.2%), *hamster* (91.8%), and *tandem bicycle* (97.7%). The hardest class is the fine-grained *kuvasz* breed, where NA-RFM obtains 30.5% top-1 accuracy and 74.2% top-5 accuracy. The top four predicted classes for this target, covering 71.1% of samples, are all visually similar large, light-colored dogs: kuvasz, malamute, Great Pyrenees, and Eskimo dog (see Figures 14 and 15). This indicates semantic steering to the intended visual category, with residual confusion among closely related breeds.

### 5.3. CelebA: Multi-Attribute Guidance

The CelebA experiment evaluates multi-attribute steering, where the target condition is a conjunction such as gender+hair or gender+age. We construct reusable guidance signals for individual attributes and combine the requested directions at inference. This setting is nontrivial because marginal attribute directions can inherit correlations from the training data; for example, 97% of blonde samples are female, so a "blond" direction can also encode gender. Implementation details are given in Appendix H.

Table 4 presents our multi-attribute guidance results. These reported stratified TFG accuracies have a higher overall average than the CelebA TFG-1 entries in Table 1, but the TFG paper does not specify them as TFG-4. We therefore compare against the reported stratified TFG scores and label the column TFG. NA-RFM is higher on **7 out of 8** attribute combinations, with average accuracy 89.7% compared with 78.8% for TFG. Two combinations reach 100.0% under the evaluation classifier (Young+Female and Old+Male), and the largest gain is on Old+Male (+30.9 percentage points).

*Table 5.* **Fine-grained bird species guidance (OOD).** Target accuracy/FID on Birds-525 species using 256 samples per species and deterministic DDIM sampling ($\eta = 0$, 100 steps).

| Species | Acc. | FID |
|---|---|---|
| Lucifer Hummingbird | **21.5%** | 24.76 |
| Scarlet Macaw | **28.1%** | 104.1 |
| Fairy Tern | **2.7%** | 93.48 |
| Brown Headed Cowbird | **3.9%** | 65.72 |
| Average (Our) | **14.1%** | 72.02 |
| TFG-4 avg. | 2.2% | 259 |

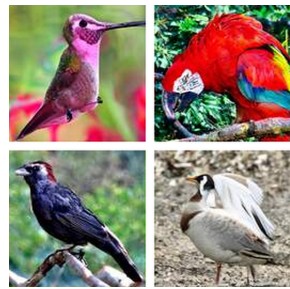

*Figure 5.* **Fine-grained bird species guidance.** Correctly classified out-of-distribution samples: Lucifer Hummingbird, Scarlet Macaw, Brown Headed Cowbird, and Fairy Tern.

TFG is only higher on Female+Non-Blond.

**Rare Combinations.** The Male+Blond combination represents only 1% of the CelebA training data. In this setting, NA-RFM reaches 68.4% accuracy, improving over TFG by 21.7 percentage points. NA-RFM remains effective even when the target combination has few training examples.

### 5.4. Fine-Grained Out-of-Distribution Guidance

The Birds-525 benchmark from Ye et al. (2024) is a hard fine-grained steering setting. We steer an ImageNet ADM model toward four bird species: three are absent from ImageNet, and Lucifer Hummingbird has only coarse overlap with the broader ImageNet hummingbird class. Table 5 reports results for four species. NA-RFM reaches 14.1% average target accuracy, compared with 2.2% for TFG-4, with the strongest results on Scarlet Macaw and Lucifer Hummingbird (28.1% and 21.5%). The absolute accuracy remains modest, as expected for fine-grained and partly out-of-distribution targets. Nevertheless, the improvement over TFG-4 and the qualitative samples in Figures 5 and 13 suggest that the learned directions carry target-species information beyond the ImageNet label set.

### 5.5. Ablation Studies

We conduct a noise level ablation study on CIFAR-10 by training RFM classifiers at $\sigma \in \{0.6, 1.0, 1.5, 2.0, 5.0\}$ and reporting RFM fit AUC on collected activations and genera-

tion accuracy. Results in Figure 6 show strong performance across $\sigma \in [0.6, 2.0]$ and degradation at $\sigma = 5.0$, supporting the use of lower-noise activations for direction discovery. Additional ablations and diagnostics are in the appendix: block selection (Appendix C), full guidance-window sweeps and timing studies (Appendix D), and direction learning ablations comparing RFM with a difference-of-means direction (Appendix E.6).

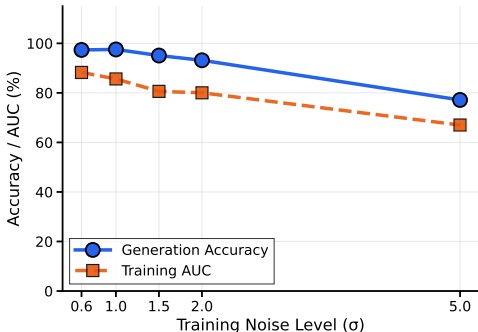

*Figure 6.* **Training noise level ablation.** Generation accuracy remains strong across $\sigma \in [0.6, 2.0]$ and degrades at $\sigma = 5.0$.

**Additional Architectures.** The main comparisons use unconditional U-Nets to match prior post-hoc guidance benchmarks. Appendix G tests the same offline/online procedure outside the main setting. On Stable Diffusion 1.5 (Rombach et al., 2022), NA-RFM works alongside text conditioning for shallow depth-of-field steering: increasing the RFM scale gives a smooth DoF sweep that often keeps the prompt-specified main subject recognizable (Figure 18); quantitatively, the foreground/background sharpness ratio increases from 2.25 to 2.58 using a Depth Anything V2-based metric (Yang et al., 2024). On transformer-based SiT-XL/2 (Ma et al., 2024), adding RFM activation steering to noise alignment raises ImageNet average accuracy from 12.9% to 61.3% and lowers FID from 220.9 to 151.8 (Table 10). Broader evaluation beyond these settings remains future work.

## 6. Discussion and Limitations

NA-RFM constructs target-specific steering directions offline from labeled examples, then samples without guidance-classifier evaluations or classifier-gradient backpropagation. Across CIFAR-10, ImageNet, CelebA, and fine-grained bird species, it gives stronger target control than gradient-based post-hoc baselines. The method avoids expensive guidance-classifier training and off-the-shelf classifier gradients, but it still requires example data for the target condition to construct the guidance signals. Broader validation for more refined conditions, more complex datasets, and more architectures remains future work.

## Acknowledgements

This material is based upon work supported by the Defense Advanced Research Projects Agency (DARPA) under Contract No. HR001125CE020, by the National Science Foundation (NSF) under grants CCF-2112665, MFAI 2502258, and MFAI 2502084, and by the Office of Naval Research (ONR) under grant N000142412631. We also gratefully acknowledge computational support provided through the NSF ACCESS program (allocation TG-CIS220009). We thank Xiao Lin and Yi Yao for helpful discussions.

## Impact Statement

This paper presents an empirical method for improving controllable generation with diffusion models. Improved control over generative models can support content creation and accessibility, while also increasing misuse risks. We encourage responsible development of both generative and detection technologies.

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

## A. A Theoretical View on RFM Direction Transfer

We consider a simplified setting that isolates one mechanism behind RFM direction transfer. The setting is a binary class-conditional Gaussian mixture with shared covariance, and the object of interest is the average gradient outer product (AGOP) of the Bayes log-odds under forward noising. In this model the shared covariance makes the log-odds affine, so its gradient is constant and the AGOP is rank one. Across noise levels, the corresponding direction changes only through a covariance-eigenvalue reweighting. This simplified model gives a local mechanism for the empirical transfer behavior: when the relevant class-separating components are reweighted similarly across the guidance window, the AGOP direction changes little with noise.

**Proposition A.1** (Direction transfer under shared covariance). *Let $y \in \{0, 1\}$ have equal class priors, and suppose*

$$\boldsymbol{x}_0 \mid y{=}k \sim \mathcal{N}(\boldsymbol{\mu}_k, \Sigma), \qquad k \in \{0, 1\},$$

*where $\Sigma$ is positive definite and $\boldsymbol{\delta} := \boldsymbol{\mu}_1 - \boldsymbol{\mu}_0 \neq 0$. Let the forward noising process be*

$$\boldsymbol{x}_t = \alpha_t \boldsymbol{x}_0 + \beta_t \boldsymbol{\epsilon}, \qquad \boldsymbol{\epsilon} \sim \mathcal{N}(\boldsymbol{0}, \boldsymbol{I}), \quad \alpha_t > 0,$$

*and define the noise-to-signal ratio $\sigma_t := \beta_t / \alpha_t$. Then the Bayes log-odds at noise level $t$ is affine in $\boldsymbol{x}_t$, and the AGOP of this log-odds is rank one. Its top eigenvector is proportional to*

$$\boldsymbol{v}^{(t)} \propto (\Sigma + \sigma_t^2 \boldsymbol{I})^{-1} \boldsymbol{\delta}.$$

*Consequently:*

1. **Isotropic covariance.** *If $\Sigma = s^2 \boldsymbol{I}$, then the normalized direction $\boldsymbol{v}^{(t)}$ is independent of $t$.*

2. **Anisotropic covariance.** *If $\Sigma = Q \mathrm{diag}(\nu_i) Q^\top$ and $\boldsymbol{\delta} = \sum_i a_i \boldsymbol{q}_i$ in the eigenbasis of $\Sigma$, then*

$$(\Sigma + \sigma_t^2 \boldsymbol{I})^{-1} \boldsymbol{\delta} = \sum_i \frac{a_i}{\nu_i + \sigma_t^2} \, \boldsymbol{q}_i.$$

*Thus changing the noise level, in this setting, only reweights the original components of the class-separation vector. Directional drift is then small when the active components of $\boldsymbol{\delta}$ lie in a spectral range that is reweighted similarly over the timesteps used for steering, and the direction varies continuously with $\sigma_t$.*

*Proof.* For equal class priors, the clean Bayes log-odds is the log-likelihood ratio. Writing

$$\log p(\boldsymbol{x}_0 \mid y{=}k) = -\frac{d}{2} \log(2\pi) - \frac{1}{2} \log \det \Sigma - \frac{1}{2} (\boldsymbol{x}_0 - \boldsymbol{\mu}_k)^\top \Sigma^{-1} (\boldsymbol{x}_0 - \boldsymbol{\mu}_k),$$

the determinant terms and the quadratic term in $\boldsymbol{x}_0$ cancel in the ratio. Thus, with

$$\ell_0(\boldsymbol{x}_0) := \log \frac{p(y{=}1 \mid \boldsymbol{x}_0)}{p(y{=}0 \mid \boldsymbol{x}_0)},$$

we obtain

$$\ell_0(\boldsymbol{x}_0) = -\frac{1}{2} (\boldsymbol{x}_0 - \boldsymbol{\mu}_1)^\top \Sigma^{-1} (\boldsymbol{x}_0 - \boldsymbol{\mu}_1) + \frac{1}{2} (\boldsymbol{x}_0 - \boldsymbol{\mu}_0)^\top \Sigma^{-1} (\boldsymbol{x}_0 - \boldsymbol{\mu}_0)$$

$$= \boldsymbol{\delta}^\top \Sigma^{-1} \left( \boldsymbol{x}_0 - \frac{\boldsymbol{\mu}_0 + \boldsymbol{\mu}_1}{2} \right).$$

Therefore

$$\nabla_{\boldsymbol{x}_0} \ell_0(\boldsymbol{x}_0) = \Sigma^{-1} \boldsymbol{\delta}, \qquad \mathbb{E}\left[ \nabla \ell_0(\boldsymbol{x}_0) \nabla \ell_0(\boldsymbol{x}_0)^\top \right] = (\Sigma^{-1} \boldsymbol{\delta})(\Sigma^{-1} \boldsymbol{\delta})^\top.$$

The Bayes-log-odds AGOP is rank one, with top eigenvector proportional to $\Sigma^{-1} \boldsymbol{\delta}$.

Under forward noising, the conditional distribution remains Gaussian:

$$\boldsymbol{x}_t \mid y{=}k \sim \mathcal{N}(\alpha_t \boldsymbol{\mu}_k, \alpha_t^2 \Sigma + \beta_t^2 \boldsymbol{I}).$$

Let

$$\Sigma_t := \alpha_t^2 \Sigma + \beta_t^2 \boldsymbol{I}, \qquad \boldsymbol{\mu}_{k,t} := \alpha_t \boldsymbol{\mu}_k, \qquad \boldsymbol{\delta}_t := \boldsymbol{\mu}_{1,t} - \boldsymbol{\mu}_{0,t} = \alpha_t \boldsymbol{\delta}.$$

Repeating the same calculation with shared covariance $\Sigma_t$ gives

$$\ell_t(\boldsymbol{x}_t) = \boldsymbol{\delta}_t^\top \Sigma_t^{-1} \left( \boldsymbol{x}_t - \frac{\boldsymbol{\mu}_{0,t} + \boldsymbol{\mu}_{1,t}}{2} \right), \qquad \nabla_{\boldsymbol{x}_t} \ell_t(\boldsymbol{x}_t) = \alpha_t \Sigma_t^{-1} \boldsymbol{\delta}.$$

Since $\alpha_t > 0$ and

$$\Sigma_t = \alpha_t^2 (\Sigma + \sigma_t^2 \boldsymbol{I}),$$

the normalized gradient direction is proportional to

$$(\Sigma + \sigma_t^2 \boldsymbol{I})^{-1} \boldsymbol{\delta}.$$

The AGOP of $\ell_t$ is again the outer product of this constant gradient with itself, so it is rank one with the same top direction. The isotropic and anisotropic statements follow by substituting $\Sigma = s^2 \boldsymbol{I}$ and by expanding $\boldsymbol{\delta}$ in the eigenbasis of $\Sigma$. $\qquad \square$

**Activation-Space Interpretation.** Proposition A.1 is stated in data space, but can also be interpreted in activation space. Suppose that, within a selected U-Net block and over the RFM guidance window, the class-conditional activations are locally approximated by a shared-covariance Gaussian mixture. The Bayes-log-odds AGOP direction then has the same form as in the proposition: a covariance-preconditioned class-mean difference whose coordinates are reweighted as the noise level changes. Direction transfer is therefore expected when the class-separating activation components remain in a stable spectral subspace over the intermediate/late window. This is the behavior measured empirically in Figure 3.

## B. Sample-Space PCA and RFM Computation

Both PCA noise alignment and RFM direction extraction operate in regimes where the ambient dimension can be much larger than the number of examples. We therefore compute the needed eigenspaces through sample-space matrices.

For PCA noise alignment, let $A_c \in \mathbb{R}^{N_c \times d_x}$ be the row-stacked image matrix for class $c$ after subtracting the class mean. We use the compact SVD

$$A_c = U_c S_c V_c^\top,$$

so the retained columns of $V_c$ are the principal directions and

$$\nu_{cj} = \frac{S_{c,jj}^2}{N_c - 1}$$

are the covariance eigenvalues used in Equation (8). This avoids forming the dense $d_x \times d_x$ image covariance matrix and is the same high-dimensional, low-sample-size computation used for the unconditional PCA statistics.

For RFM, let $G \in \mathbb{R}^{N \times d_h}$ stack the RFM gradient features $\nabla_{\boldsymbol{h}} f(\boldsymbol{h}_i)^\top$ as rows. The activation-space AGOP matrix is

$$A = \frac{1}{N} G^\top G \in \mathbb{R}^{d_h \times d_h}.$$

In our regime $d_h = C_\ell H_\ell W_\ell$ is large and $N \ll d_h$, so we instead diagonalize the sample-space matrix

$$B = \frac{1}{N} GG^\top \in \mathbb{R}^{N \times N}.$$

If $B\boldsymbol{a}_j = \rho_j \boldsymbol{a}_j$ with $\rho_j > 0$ and $\|\boldsymbol{a}_j\|_2 = 1$, define

$$\boldsymbol{u}_j = \frac{G^\top \boldsymbol{a}_j}{\sqrt{N\rho_j}}.$$

Then $\|\boldsymbol{u}_j\|_2 = 1$ and

$$A\boldsymbol{u}_j = \frac{1}{N} G^\top G \frac{G^\top \boldsymbol{a}_j}{\sqrt{N\rho_j}} = \frac{1}{\sqrt{N\rho_j}} G^\top \left( \frac{1}{N} GG^\top \boldsymbol{a}_j \right)$$

$$= \rho_j \frac{G^\top \boldsymbol{a}_j}{\sqrt{N\rho_j}} = \rho_j \boldsymbol{u}_j.$$

Thus every positive-eigenvalue eigenvector of the sample-space matrix gives the corresponding nonzero eigenvector of the activation-space AGOP. The zero eigenspace of $A$ is irrelevant for direction discovery because it contains directions orthogonal to all RFM gradient features.

## C. Block Selection Ablation

We compare NA-RFM guidance using features from three U-Net locations: Encoder-9 (8×8 resolution), Middle block (4×4 bottleneck), and Decoder-6 (8×8 resolution), using 256 samples per CIFAR-10 class to isolate the effect of block choice.

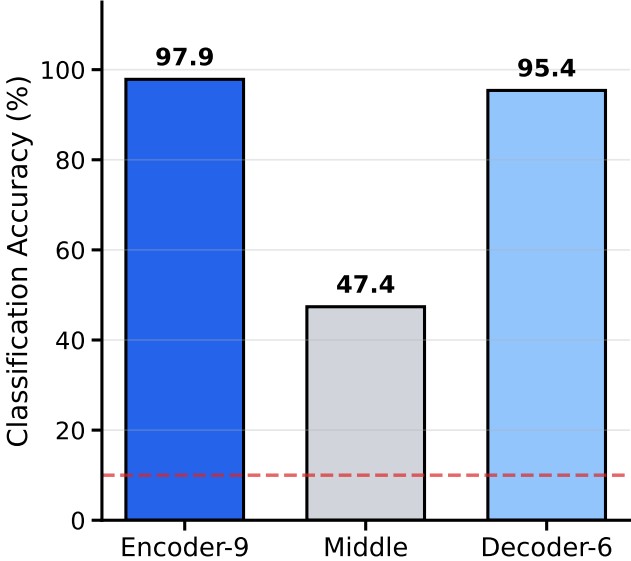

*Figure 7.* **Block selection ablation.** NA-RFM guidance accuracy on CIFAR-10 (256 samples per class) for three U-Net block locations. Encoder-9 achieves highest accuracy (97.9%), Decoder-6 is comparable (95.4%), and the Middle block is lower in this CIFAR-10 setting (47.4%). Red dashed line indicates random chance (10%).

As shown in Figure 7, both encoder and decoder blocks at 8×8 resolution achieve high accuracy (97.9% and 95.4% respectively), while the middle bottleneck block is lower in this CIFAR-10 ablation (47.4%). For CIFAR-10, these results favor an intermediate spatial resolution: the block retains more spatial detail than the bottleneck while still carrying class-relevant features.

The CIFAR-10 gap is consistent with two structural properties of the U-Net bottleneck. (1) *Limited spatial resolution.* The middle block operates at the network's spatial bottleneck ($4 \times 4$ for $32 \times 32$ inputs), where spatial information is highly compressed; a single steering direction there has less spatial capacity than in the neighboring $8 \times 8$ blocks. (2) *Skip connections bypass the bottleneck.* Decoder blocks combine upsampled middle-block features with encoder features through skip connections, so an edit applied inside the bottleneck can be partly diluted by unedited encoder features on the way out. Editing at the last encoder block before the bottleneck propagates through both the bottleneck and the skip path, giving the steering direction two downstream routes. This explanation is setting-dependent.

## D. Full Guidance Window Ablation

We test how the RFM guidance window affects generation. Although each RFM direction is trained at one reference noise level, sampling can apply the same direction over a wider range of timesteps.

Figure 8 presents a timing-window ablation with **RFM-only** guidance on CIFAR-10 with 256 samples per class. Steering in the second half (steps 50–99) reaches 90.2% accuracy, close to the full-window result, whereas steering in the first half (steps 0–49) reaches 24.2%. Steering only near the RFM training noise level gives 13.4% accuracy. Accuracy increases as the late guidance window grows: 10 steps gives 35.2%, 20 steps gives 60.8%, 30 steps gives 75.6%, and 50 steps gives 90.2%.

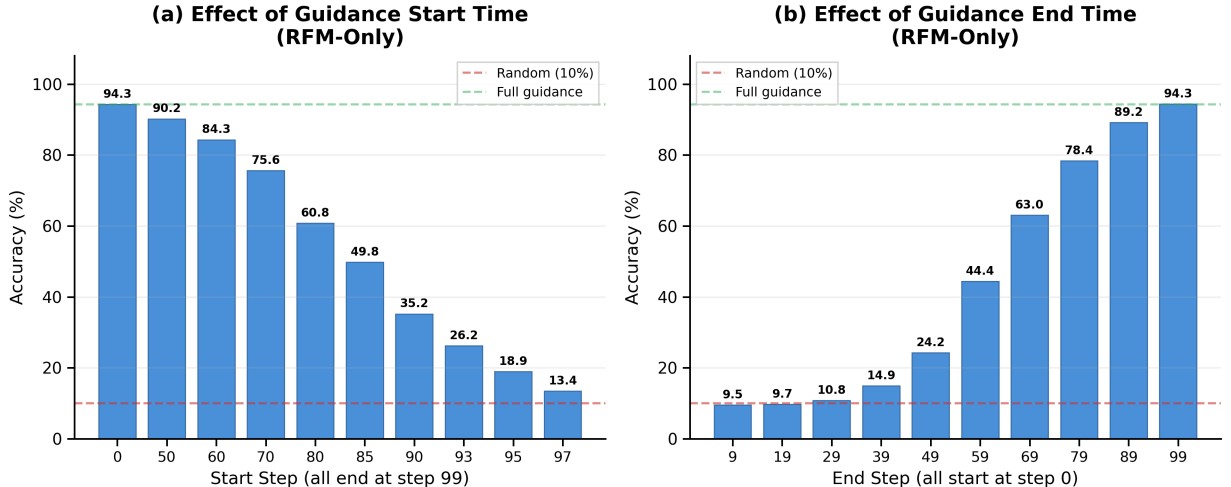

*Figure 8.* **RFM-only timing window ablation.** We vary which DDIM steps receive RFM guidance. *(a) Effect of guidance start time –* end step fixed at 99, start swept from 0 to 97: accuracy decreases from 94.3% (full window) to 13.4% when steering is restricted to the narrow band near the RFM training noise level. *(b) Effect of guidance end time –* start fixed at 0, end swept from 9 to 99: accuracy rises from 9.5% (early steps only) to 94.3% (full window). The two panels indicate that, in this CIFAR-10 setting, RFM steering benefits from a sufficiently long late-stage window.

## E. Additional Experimental Results

### E.1. CIFAR-10 Per-Class Analysis

Table 6 reports the per-class accuracy and FID behind the CIFAR-10 average in Table 2. Accuracy remains high across the ten classes, while per-class FID ranges from 22.5 (automobile) to 62.3 (airplane).

*Table 6.* **CIFAR-10 per-class results.** Per-class accuracy and FID for the reported NA-RFM setting.

| Class | Accuracy | FID |
|---|---|---|
| airplane | 95.5% | 62.3 |
| automobile | 98.1% | 22.5 |
| bird | 96.2% | 54.9 |
| cat | 97.2% | 50.7 |
| deer | 98.4% | 38.5 |
| dog | 92.1% | 44.9 |
| frog | 94.6% | 44.3 |
| horse | 99.4% | 36.3 |
| ship | 96.9% | 35.6 |
| truck | 97.4% | 24.3 |
| **Average** | **96.6%** | **41.4** |

### E.2. CIFAR-10 Samples

These fixed-seed grids show how the same noise draws change under noise alignment and under the full NA-RFM sampler. They provide visual context for the CIFAR-10 results in Tables 2 and 6; the quantitative accuracy and FID are reported in the tables.

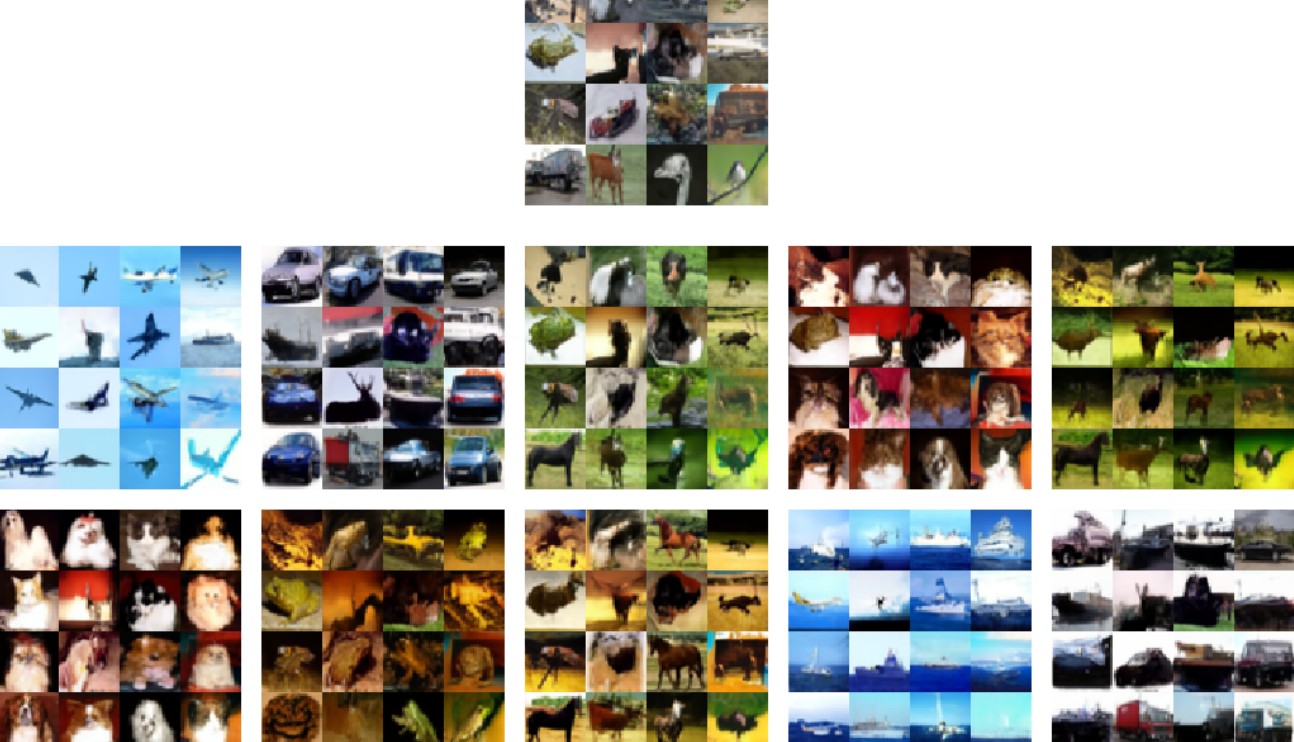

*Figure 9.* **Fixed-seed CIFAR-10: unguided and noise alignment.** Each panel shows the first 16 samples from the shared 64-sample fixed-seed bank. The top panel is unguided; the guided panels use noise alignment ($\lambda = 8$), ordered as airplane, automobile, bird, cat, deer, dog, frog, horse, ship, and truck.

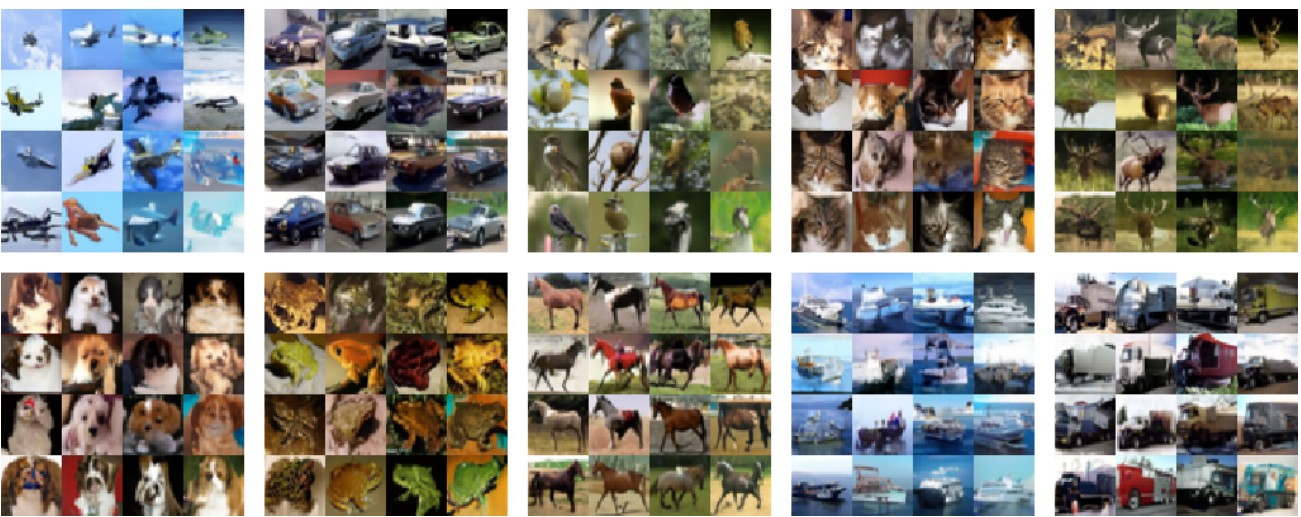

*Figure 10.* **Fixed-seed CIFAR-10: NA-RFM.** Each panel shows the same first 16 fixed-seed samples using the reported NA-RFM setting ($\lambda = 3$, $w_{\mathrm{RFM}} = 1$, $s = 2$), ordered as airplane, automobile, bird, cat, deer, dog, frog, horse, ship, and truck.

### E.3. CelebA Multi-Attribute Guidance Samples

We visualize the two multi-attribute CelebA settings reported in Table 4. Each panel corresponds to a conjunction of two attributes, and the percentages are measured by the same attribute classifiers used for the table.

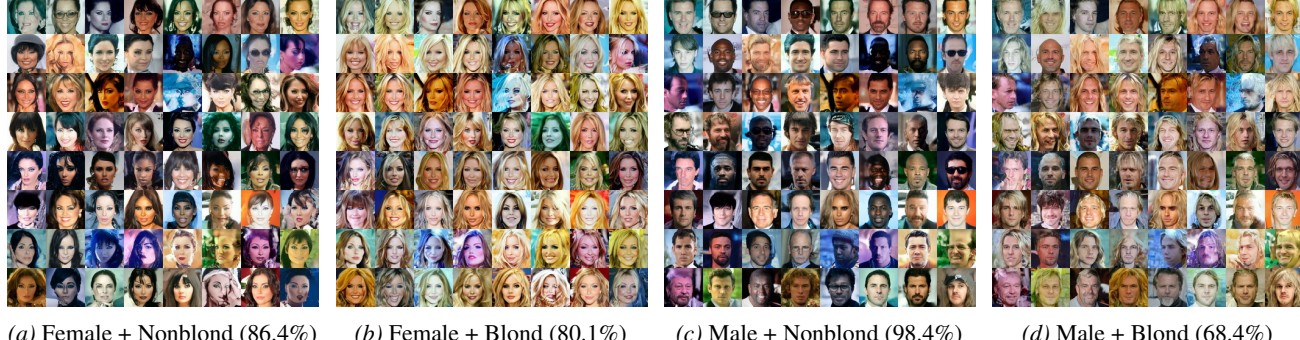

(a) Female + Nonblond (86.4%)    (b) Female + Blond (80.1%)    (c) Male + Nonblond (98.4%)    (d) Male + Blond (68.4%)

*Figure 11.* **CelebA Gender+Hair guidance results.** Each panel shows 64 samples generated with the corresponding combined attribute guidance. Accuracy indicates the fraction classified correctly for both attributes. Our method achieves 83.3% average accuracy vs. TFG's 75.4%.

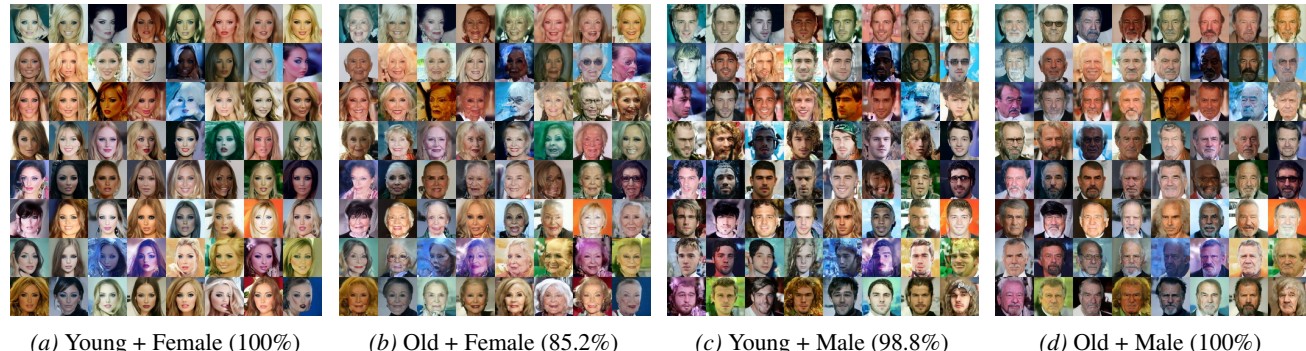

(a) Young + Female (100%)    (b) Old + Female (85.2%)    (c) Young + Male (98.8%)    (d) Old + Male (100%)

*Figure 12.* **CelebA Gender+Age guidance results.** Each panel shows 64 samples generated with the corresponding combined attribute guidance. Accuracy indicates the fraction classified correctly for both attributes. Our method achieves 96.0% average accuracy vs. TFG's 82.3%.

### E.4. Fine-Grained Bird Species Guidance Samples

The following grids show samples for the four bird targets in Table 5. The percentages in the subcaptions are exact species-level accuracy under the Birds-525 classifier; visually similar but different species count as errors in this metric.

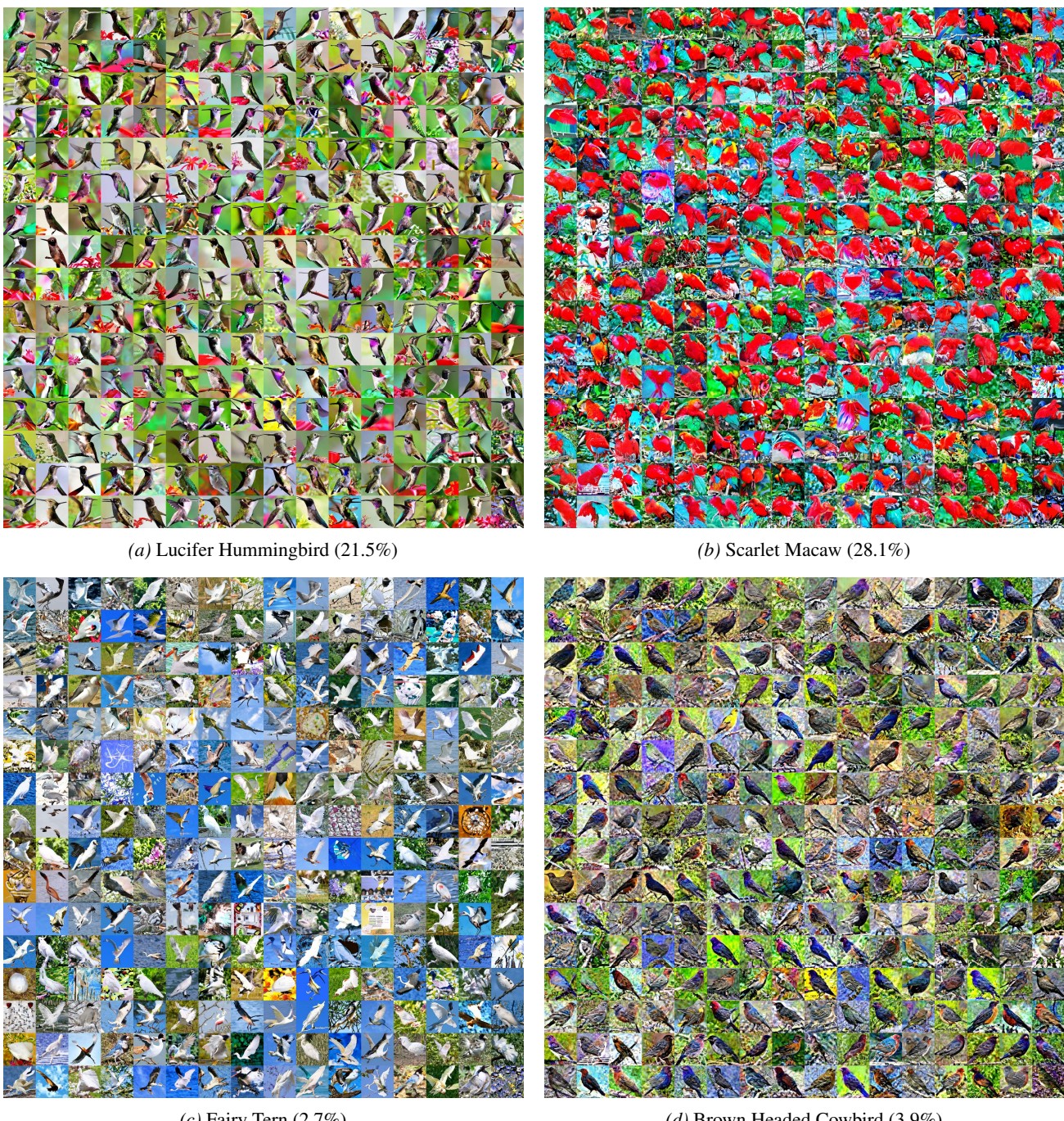

*(a)* Lucifer Hummingbird (21.5%)    *(b)* Scarlet Macaw (28.1%)

*(c)* Fairy Tern (2.7%)    *(d)* Brown Headed Cowbird (3.9%)

*Figure 13.* **Fine-grained bird species generation samples (16×16 grid per species).** Generated using deterministic DDIM sampling ($\eta = 0$, 100 DDIM sampling steps). Percentages indicate target species accuracy evaluated on 256 samples per species using the Birds-525 classifier. The grids provide qualitative context for Table 5: for species absent from ImageNet, samples are often bird-like but may be classified as nearby species rather than the exact target.

## E.5. ImageNet Label Guidance Visualizations

We include qualitative ImageNet samples and classifier-confusion statistics for the four targets in Table 1. These figures are intended to complement the quantitative accuracy and FID numbers, not to replace them.

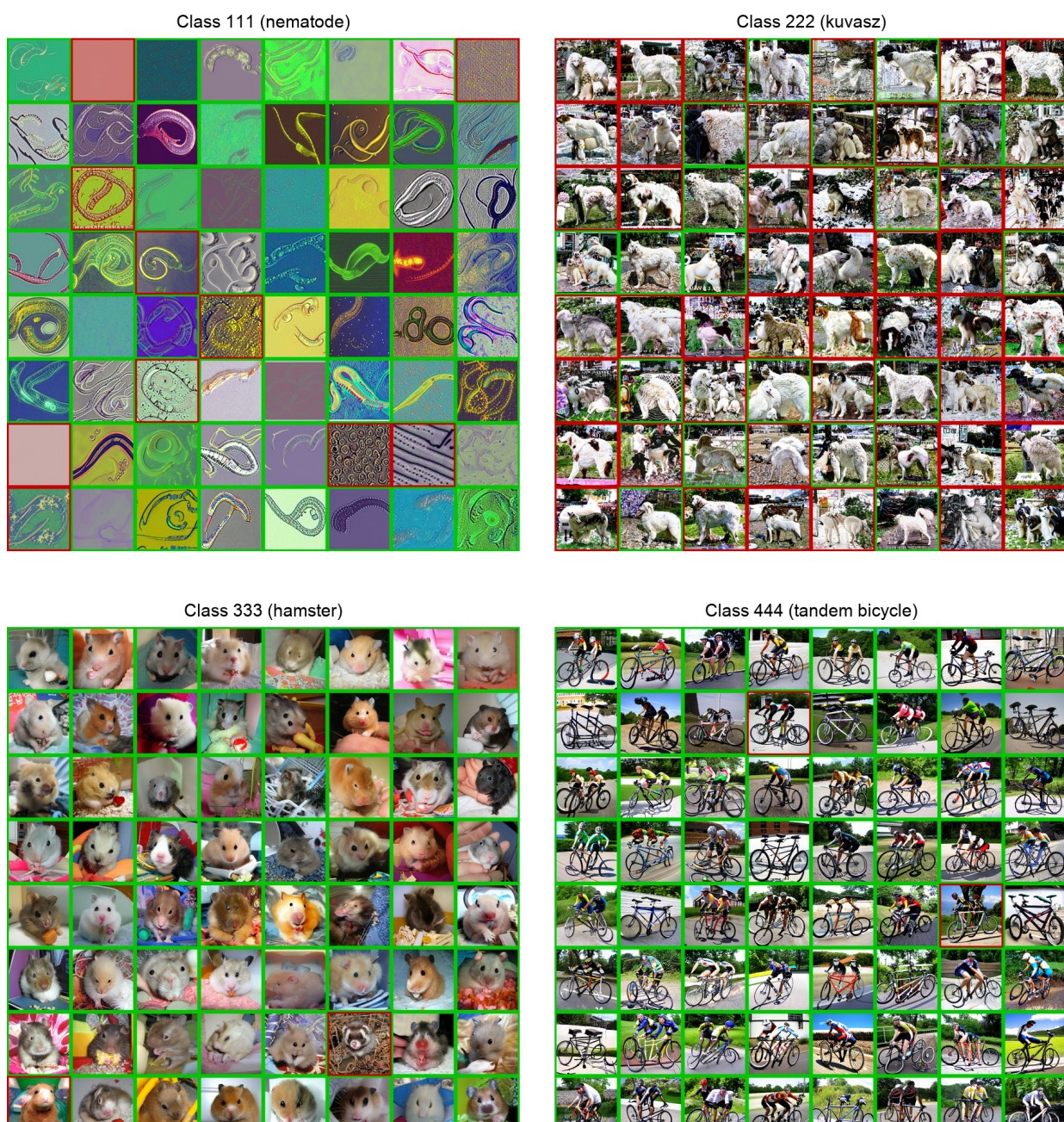

*Figure 14.* **ImageNet label guidance sample grid.** Generated samples for four target classes (nematode, kuvasz, hamster, and tandem bicycle) using the settings reported in Table 19. Each class shows an 8×8 qualitative grid; green/red borders indicate correct/incorrect evaluation-classifier predictions. Per-class metrics are discussed in Section 5.2 and reported in Table 19; aggregate averages are in Table 1.

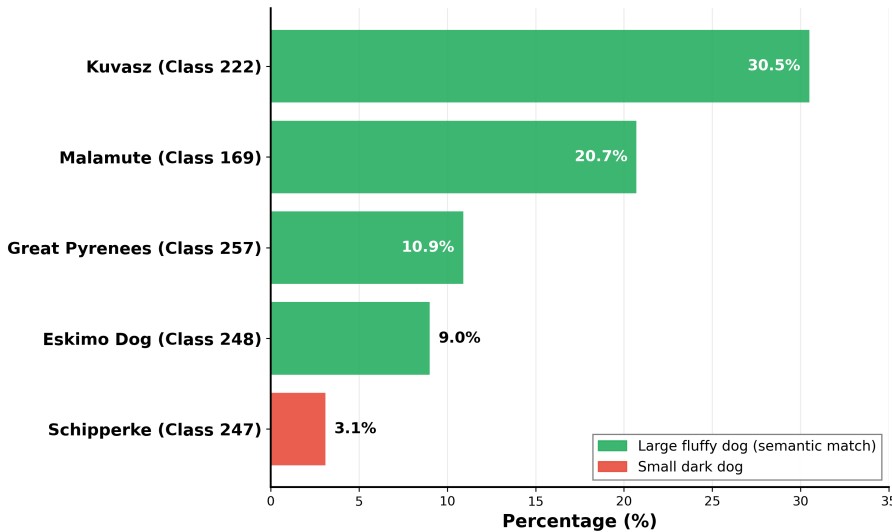

*Figure 15.* **Class 222 (kuvasz) confusion analysis.** The leading predictions concentrate on large, light-colored dog breeds: kuvasz (30.5%), malamute (20.7%), Great Pyrenees (10.9%), and Eskimo dog (9.0%). The resulting top-5 accuracy is 74.2%, indicating that many errors remain within a visually similar breed cluster even though exact top-1 kuvasz accuracy is lower.

### E.6. Direction Learning Ablation: Difference-of-Means vs. RFM

We compare the learned RFM direction against a simpler baseline: a difference-of-means direction $\mathbf{d} = \mathbb{E}[\mathbf{h}|y = c] - \mathbb{E}[\mathbf{h}]$, where $\mathbf{h}$ denotes intermediate activations. This baseline tests whether the difference-of-means direction is sufficient for activation steering.

*Table 7.* **Direction learning ablation on ImageNet kuvasz (class 222).** On this target, the RFM direction gives substantially higher top-1 accuracy than the difference-of-means direction. The last column summarizes the qualitative behavior visible in Figure 16.

| Method | Accuracy | Label Confidence | Qualitative behavior |
|---|---|---|---|
| Difference-of-means direction | 6.2% | 0.08 | Repeated standing profiles |
| **RFM direction for activation steering (Ours)** | **30.5%** | **0.24** | More varied dog poses |

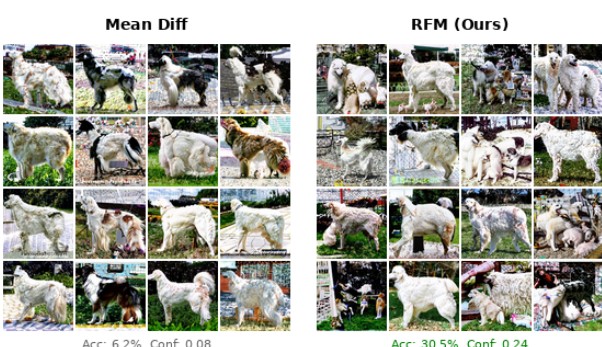

*Figure 16.* **Difference-of-means vs. RFM direction comparison on kuvasz (class 222).** *Left:* the difference-of-means direction produces many samples with similar standing dog profiles. *Right:* RFM guidance yields higher top-1 accuracy and a visibly broader set of poses in this grid.

On this target, the difference-of-means direction is weak: it gives 6.2% top-1 accuracy and the displayed samples share a narrow standing-dog profile. The RFM direction, learned from the target-vs-rest AGOP eigenspace, gives 30.5% top-1 accuracy and produces a visibly broader set of poses in the same grid. We therefore use RFM, rather than the difference-of-means direction, for direction discovery. This is consistent with the simplified analysis in Appendix A, where the AGOP

direction in a shared-covariance Gaussian model follows a covariance-weighted class-separation direction that remains stable under forward noising.

## F. Diversity Analysis and Classifier Calibration

A natural concern for any guidance method that drives accuracy upward is whether this comes at the cost of *diversity*–for example, whether the samples concentrate on a small set of easy-to-classify images. A related concern is whether the reported accuracy gains depend on the specific evaluation classifier. We address both here.

### F.1. Diversity: Generative Recall

Beyond FID, we measure diversity using the precision/recall metric of Kynkäänniemi et al. (2019), where recall estimates the fraction of the reference (real) distribution covered by the generated samples. We compare NA-RFM against the same training-free TFG-4 baseline and the noise-conditioned classifier-guidance baseline on CIFAR-10, using a ConvNeXt-Tiny feature extractor on 2,048 generated samples per class.

*Table 8.* **Accuracy, FID, and generative recall on CIFAR-10 label guidance.** NA-RFM achieves the highest accuracy, the lowest FID, and the highest recall simultaneously. Recall is computed following Kynkäänniemi et al. (2019).

| Method | Accuracy ↑ | FID ↓ | Recall ↑ |
|---|---|---|---|
| TFG-4 (Ye et al., 2024) | 77.1% | 73.9 | 0.362 |
| Classifier Guidance (Nichol & Dhariwal, 2021) | 86.0% | 41.9 | 0.430 |
| **NA-RFM (Ours)** | **96.6%** | **41.4** | **0.442** |

Table 8 shows that the accuracy gain is not accompanied by lower recall in this comparison: NA-RFM achieves the highest guidance accuracy and the highest recall among the three methods, while also improving FID. Its recall is 22% higher than the gradient-based TFG-4 baseline in relative terms and slightly exceeds that of noise-conditioned classifier guidance, indicating that the accuracy gain does not reduce coverage under this metric.

### F.2. Robustness to the Evaluation Classifier

To assess evaluator dependence, we re-evaluate the same CIFAR-10 and ImageNet generations with additional, architecturally different classifiers.

*Table 9.* **NA-RFM accuracy under different evaluation classifiers.** Architectures span CNNs (ResNet56, ResNet-50, VGG19-BN, ConvNeXt-Base) and vision transformers (ConvNeXt-Tiny, DeiT-Small). The results are within $\pm 1$–4 points of one another on both CIFAR-10 and ImageNet, suggesting that the gains reported in Table 1 are not specific to one evaluator.

| Benchmark | Evaluation classifier | NA-RFM Accuracy |
|---|---|---|
| CIFAR-10 | ConvNeXt-Tiny (reported in main) | 96.6% |
| | ResNet56 | 96.5% |
| | VGG19-BN | 97.2% |
| ImageNet | DeiT-Small (Touvron et al., 2021) (reported) | 75.8% |
| | ResNet-50 | 71.8% |
| | ConvNeXt-Base (Liu et al., 2022) | 75.5% |

As shown in Table 9, NA-RFM's accuracy is stable across classifiers: on CIFAR-10 we obtain 96.5–97.2% across ConvNeXt-Tiny, ResNet56, and VGG19-BN; on ImageNet we obtain 71.8–75.8% across DeiT-Small, ResNet-50, and ConvNeXt-Base. The main Table 1 uses the same evaluation classifiers as the TFG baseline (Ye et al., 2024) to ensure a fair like-for-like comparison.

## G. Extending NA-RFM Beyond Unconditional U-Nets

The main body of the paper evaluates NA-RFM on unconditional U-Net diffusion models. Here we include two extensions: a *conditional* text-to-image model (Stable Diffusion 1.5 (Rombach et al., 2022)), where the target visual attribute is supplied

through examples rather than as a built-in text condition, and a *transformer-based* latent diffusion model (SiT-XL/2 (Ma et al., 2024)), where the U-Net inductive bias is absent. Both experiments keep the offline direction-learning and online activation-steering structure of the main experiments.

### G.1. Steering Transformer-Based Latent Diffusion: SiT-XL/2

We apply NA-RFM to the official pretrained SiT-XL/2 (Ma et al., 2024), a transformer-based latent diffusion model with 28 transformer blocks built on the DiT architecture (Peebles & Xie, 2023), using null-class conditioning as the unconditional baseline. We use a small class-specific set of middle transformer blocks; the row-wise sampling-time steering settings are reported separately in Table 11, and systematic transformer-specific block selection is left to future work.

*Table 10.* **NA-RFM on SiT-XL/2 (transformer-based latent diffusion).** We report per-class top-1 accuracy and FID for NA-RFM and for noise-alignment-only (PCA-based noise alignment without RFM activation steering), on the same 4 ImageNet classes as in the main body (256 samples per class). Adding RFM activation steering on top of noise alignment raises average accuracy from 12.9% to 61.3% and lowers FID from 220.9 to 151.8. The U-Net-based ADM TFG-4 baseline reports 59.8% average accuracy.

| | NA-RFM (NA+RFM) | | Noise-align only | |
| --- | --- | --- | --- | --- |
| **Class** | **Acc. ↑** | **FID ↓** | **Acc. ↑** | **FID ↓** |
| 111 (nematode) | 52.7% | 202.4 | 46.9% | 197.0 |
| 222 (kuvasz) | 28.5% | 172.3 | 2.7% | 197.3 |
| 333 (hamster) | 81.6% | 129.1 | 1.6% | 227.7 |
| 444 (tandem bicycle) | 82.4% | 103.5 | 0.4% | 261.8 |
| **Average** | **61.3%** | **151.8** | 12.9% | 220.9 |

*Table 11.* **SiT-XL/2 row-wise sampling-time steering settings for Table 10.** Noise alignment is active for $\sigma_t > \sigma_{\text{end}}$ in the model's native Karras/EDM-style schedule; RFM steering is active on the listed native-$\sigma$ interval. The RFM coefficient is applied to each steered transformer block. Common settings are 100 sampling steps, null-class conditioning as the unconditional branch, and 256 samples per class.

| Class | $\lambda$ | $w_{\text{RFM}}$ | $s$ | $\sigma_{\text{end}}$ | $\sigma_t \in [\sigma_{\text{R}}^{\text{lo}}, \sigma_{\text{R}}^{\text{hi}}]$ | RFM steering blocks |
| --- | --- | --- | --- | --- | --- | --- |
| 111 (nematode) | 2.0 | 0.03/block | 6.0 | 25 | $[0.0026, 17.53]$ | 16, 20 |
| 222 (kuvasz) | 3.0 | 0.025/block | 4.0 | 40 | $[0.0026, 5.84]$ | 12, 16, 20 |
| 333 (hamster) | 2.0 | 0.03/block | 2.0 | 40 | $[0.0026, 5.84]$ | 16, 18, 20 |
| 444 (tandem bicycle) | 3.0 | 0.04/block | 6.0 | 15 | $[0.0026, 17.53]$ | 16, 20 |

Table 10 reports the same two components on SiT-XL/2. Noise alignment alone reaches 46.9% on nematode but is weak on the other three targets. Adding RFM activation steering raises average accuracy from 12.9% to 61.3% and improves FID from 220.9 to 151.8. These numbers place the SiT-XL/2 run in the same accuracy range as the U-Net TFG-4 baseline reported in Ye et al. (2024), while using a different backbone and a small class-specific block set. Qualitative samples are shown in Figure 17.

These extensions keep the implementation structure close to the U-Net experiments: choose a layer, learn an additive direction from examples, and apply that direction during sampling.

### G.2. Steering Conditional Models: Depth-of-Field on Stable Diffusion 1.5

**Motivation.** A text-conditional diffusion model like Stable Diffusion 1.5 is trained on (image, caption) pairs, so it can respond to caption-level conditions but does not provide direct controls for purely visual attributes that are not routinely described in captions. Depth-of-field (DoF) – how sharply the foreground is separated from a blurred background – is one such attribute. We use it to test whether NA-RFM can steer a conditional model toward a visual attribute specified by examples rather than by the prompt interface.

**Direction Learning.** We use the `svnfs/depth-of-field` dataset from Hugging Face, which provides binary shallow/deep DoF labels across approximately 600 real images per class. We extract activations from `unet.down_blocks[2].resnets[-1]` of the SD 1.5 U-Net at forward-noise level $\sigma \approx 0.344$, flattening the

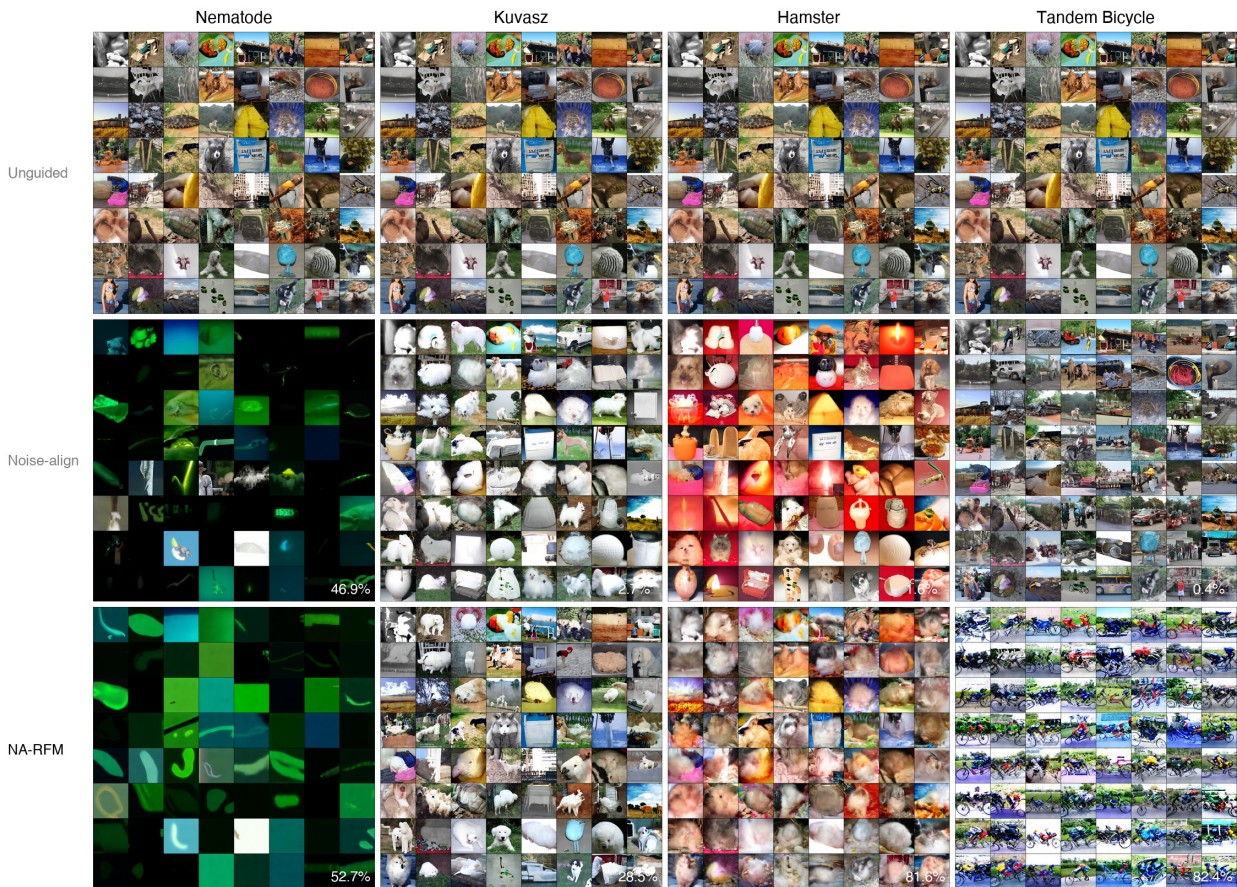

*Figure 17.* **NA-RFM on SiT-XL/2.** Columns correspond to the four ImageNet targets; rows show unguided sampling, noise-alignment-only guidance, and full NA-RFM. Adding RFM activation steering gives more recognizable samples for *kuvasz*, *hamster*, and *tandem bicycle* in this qualitative grid.

activations into a 327,680-dimensional representation. We then train an RFM direction with bandwidth 5000 on the shallow-vs.-deep labels.

**Evaluation Metric.** We estimate per-pixel depth with Depth Anything V2 (Yang et al., 2024), segment each generated image into foreground (top 40% depth) and background (bottom 60%), compute the Laplacian variance on each region as a sharpness proxy, and report the foreground-to-background sharpness ratio. A higher ratio indicates a sharper foreground against a blurred background, matching the shallow-DoF target.

**Setup and Results.** We take the first 25 COCO-Karpathy validation prompts and generate 4 images per prompt (100 images total) with and without NA-RFM steering. Steering *toward* shallow DoF produces, on average:

- background sharpness reduced by **27.6%**;
- foreground-to-background sharpness ratio increased from **2.25** (unsteered) to **2.58** (steered).

The quantitative run uses 50 DDIM sampling steps over $0.041 \leq \sigma_t \leq 13.12$, text CFG scale 7.5, RFM coefficient $w_{\mathrm{RFM}} = 0.7$, and RFM amplification scale $s = 4.0$ in the three-branch prediction

$$\epsilon = \epsilon_u + 7.5(\epsilon_c - \epsilon_u) + 4.0(\epsilon_s - \epsilon_c),$$

where $\epsilon_s$ is the conditional prediction with the RFM steering layer active. The coefficient 7.5 is the text-CFG scale, while the last term is the separate RFM amplification term. The four seeds are 42, 123, 256, and 789 for each prompt. Across prompts, steering often keeps the main subject recognizable while increasing background blur; Figure 18 shows qualitative

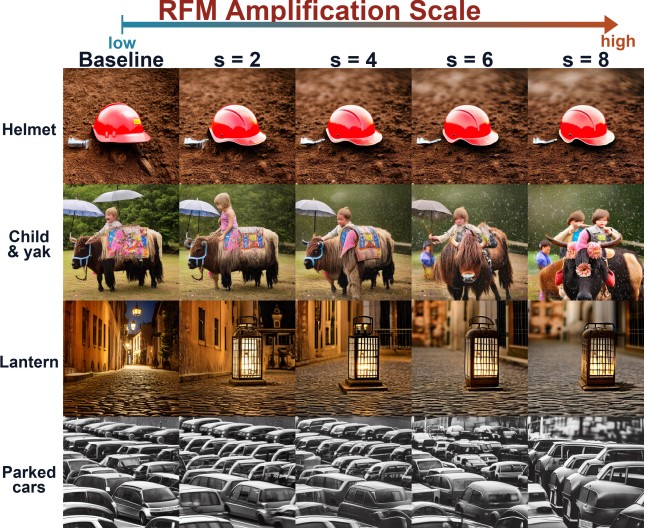

*Figure 18.* **Depth-of-field steering on Stable Diffusion 1.5.** For each prompt (rows), we sweep the NA-RFM steering strength from $0$ (left, baseline) to its maximum (right). The displayed sweep uses increasing RFM amplification scale $s$. As the strength increases, the foreground subject often remains recognizable while the background becomes progressively more blurred, consistent with a shallow-DoF target.

examples. These results suggest that NA-RFM can steer a *conditional* model toward an example-defined visual attribute, without retraining and without inference-time gradients.

This experiment shows that NA-RFM can learn guidance signals from examples for those visual properties that are difficult to control precisely with prompts.

## H. Implementation Details

We provide detailed implementation specifications for reproducibility. Unless otherwise noted, the main U-Net experiments of NA-RFM use deterministic DDIM sampling with 100 sampling steps; RFM-active steps add a second steered denoiser forward pass.

We report effective guidance parameters throughout: $\lambda$ is the noise-alignment coefficient, $w_{\mathrm{RFM}}$ is the RFM steering coefficient, $s$ is the RFM amplification scale, and inactive components are denoted by 0 or "–".

### H.1. Noise-Level Reporting

We report guidance windows by the noise parameter $\sigma_t$ from Equation (2). For VP/DDPM-style schedules this is the noise-to-signal ratio

$$\sigma_t = \sqrt{\frac{1 - \bar{\alpha}_t}{\bar{\alpha}_t}},$$

where $\bar{\alpha}_t$ is the cumulative product of the per-step VP signal factors; equivalently, with $\bar{\beta}_t := 1 - \bar{\alpha}_t$, $\sigma_t = \sqrt{\bar{\beta}_t/\bar{\alpha}_t}$. Thus the reporting convention applies to any VP/DDPM variance schedule, not only a particular choice. For EDM/Karras-style samplers, $\sigma_t$ is the scheduler's native noise level. For SiT/DiT-style runs, we report the native noise variable used by that scheduler rather than converting it with the VP/DDPM $\bar{\alpha}_t$ formula. We report noise alignment by the cutoff $\sigma_{\mathrm{end}}$, active when $\sigma_t \geq \sigma_{\mathrm{end}}$, and RFM steering by the interval $\sigma_t \in [\sigma_{\mathrm{R}}^{\mathrm{lo}}, \sigma_{\mathrm{R}}^{\mathrm{hi}}]$. CIFAR-10 and the ADM ImageNet/Birds runs use a VP/DDPM schedule with $T = 1000$ training timesteps and linear per-step variances $\delta_i \in [0.0001, 0.02]$; the CIFAR-10 RFM collection level is $\sigma \approx 0.21$, $\sigma_{\mathrm{end}} = 3.33$, and the RFM window corresponding to steps 30–99 is $[0.01, 11.69]$. For Birds-525, RFM steering is applied over the full 100-step DDIM window, reported as $[0.01, 157.4]$. The Stable Diffusion 1.5 DoF run uses a scaled-linear variance schedule with 50 DDIM sampling steps; its direction-extraction level is $\sigma \approx 0.344$ and its full RFM window is $[0.041, 13.12]$. CelebA-HQ and ImageNet rows in the implementation tables below are already specified directly in $\sigma$ units.

## H.2. Model Checkpoints and Architecture

*Table 12.* **Diffusion model checkpoints and architectures.**

| Dataset | Architecture | Checkpoint URL |
|---------|--------------|----------------|
| CIFAR-10 | Improved DDPM U-Net | `https://openaipublic.blob.core.windows.net/diffusion/march-2021-ema/cifar10_uncond_50M_500K.pt` |
| ImageNet | ADM U-Net (256×256) | `https://openaipublic.blob.core.windows.net/diffusion/jul-2021/256x256_diffusion_uncond.pt` |
| CelebA-HQ | DDPM U-Net | `https://huggingface.co/google/ddpm-ema-celebahq-256` |

The ImageNet ADM architecture uses: 256 base channels, channel multipliers [1, 1, 2, 2, 4, 4], 2 residual blocks per resolution, attention at 32×32, 16×16, and 8×8 resolutions, 64 channels per attention head, and learned sigma prediction.

## H.3. Guidance and Evaluation Classifiers

*Table 13.* **Auxiliary classifiers** used for pseudo-labeling, baseline compatibility, or guidance/evaluation protocols.

| Dataset | Architecture | Source |
|---------|--------------|--------|
| CIFAR-10 | ResNet-18 | OpenOOD benchmark |
| ImageNet | ViT-B/16 | torchvision pretrained |
| CelebA (Age) | ViT | `https://huggingface.co/nateraw/vit-age-classifier` |
| CelebA (Gender) | ViT | `https://huggingface.co/rizvandwiki/gender-classification-2` |
| CelebA (Hair) | ViT | `https://huggingface.co/enzostvs/hair-color` |
| Birds-525 | EfficientNet | `https://huggingface.co/chriamue/bird-species-classifier` |

*Table 14.* **Evaluation classifiers** (used for evaluating guidance accuracy).

| Dataset | Architecture | Source |
|---------|--------------|--------|
| CIFAR-10 | ConvNeXT-Tiny | `https://huggingface.co/ahsanjavid/convnext-tiny-finetuned-cifar10` |
| ImageNet | DeiT-Small | `https://huggingface.co/facebook/deit-small-patch16-224` |
| CelebA (Age) | Swin | `https://huggingface.co/ibombonato/swin-age-classifier` |
| CelebA (Gender) | ViT | `https://huggingface.co/rizvandwiki/gender-classification` |
| CelebA (Hair) | ViT | `https://huggingface.co/londe33/hair_v02` |
| Birds-525 | EfficientNet-B2 | `https://huggingface.co/dennisjooo/Birds-Classifier-EfficientNetB2` |

## H.4. Training Data for Direction Discovery

*Table 15.* **Training data** used for computing PCA statistics and RFM directions.

| Dataset | Size | Source |
|---|---|---|
| CIFAR-10 | 50,000 images | `torchvision.datasets.CIFAR10` |
| ImageNet-1k | 1.28M images | https://huggingface.co/datasets/imagenet-1k |
| CelebA-HQ | 30,000 images | https://github.com/tkarras/progressive_growing_of_gans |
| Birds-525 | 89,885 images | https://huggingface.co/datasets/chriamue/bird-species-dataset |

## H.5. CIFAR-10 Implementation

*Table 16.* **CIFAR-10 implementation details.**

| Parameter | Value |
|---|---|
| *Diffusion Model* | |
| Architecture | OpenAI U-Net (improved DDPM) |
| Image resolution | 32×32 |
| Noise schedule | Linear $\beta \in [0.0001, 0.02]$, 1000 training timesteps |
| *Activation Collection* | |
| RFM steering layer | `input_blocks_9` |
| Feature map resolution | 8×8 |
| Collection noise level | $\sigma \approx 0.21$ |
| Samples per class | 1,000 |
| Training data | CIFAR-10 train split (50,000 images) |
| *RFM Training* | |
| Kernel | Laplace |
| Bandwidth | 100 |
| Regularization | $10^{-3}$ |
| Iterations | 5 |
| Top-$k$ eigenvectors | 3 |
| *Guidance Settings* | |
| RFM amplification scale $s$ | 2.0 |
| RFM window $\sigma_t \in [\sigma_R^{lo}, \sigma_R^{hi}]$ | $[0.01, 11.69]$ |
| RFM coefficient $w_{RFM}$ | 1.0 |
| Noise-alignment cutoff $\sigma_{end}$ | 3.33 |
| Noise alignment coefficient $\lambda$ | 3.0 |

*Table 17.* **CIFAR-10 guidance settings used for the noise-alignment and classifier-guidance comparisons.** The NA-RFM row is repeated only to make the comparison with the noise-alignment-only and classifier-guidance baselines explicit.

| Method | $\lambda$ | $w_{RFM}$ | $s$ | Sampler / guidance setting | Reported metrics |
|---|---|---|---|---|---|
| Noise alignment only | 8.0 | 0 | – | DDIM, $\eta = 0$; $\sigma_{end} = 3.33$; no RFM steering | 80.0% acc., 120 FID |
| NA-RFM | 3.0 | 1.0 | 2.0 | DDIM, $\eta = 0$; $\sigma_{end} = 3.33$; RFM window $[0.01, 11.69]$ at `input_blocks_9` | 96.6% acc., 41.4 FID |
| Classifier guidance | – | – | – | TFG CIFAR-10 script; classifier-guidance coefficient 5.0; DDIM, $\eta = 1$ | 86.0% acc. |

## H.6. ImageNet Implementation

*Table 18.* **ImageNet implementation details.**

| Parameter | Value |
| --- | --- |
| *Diffusion Model* | |
| Architecture | ADM U-Net (unconditional) |
| Image resolution | 256×256 |
| Noise schedule | Linear $\beta \in [0.0001, 0.02]$, 1000 training timesteps |
| *Activation Collection* | |
| RFM steering layer | `input_blocks_15` |
| Feature map resolution | 8×8 |
| Feature channels | 1024 |
| Collection noise level | $\sigma \approx 0.59$ |
| Training data | ImageNet-1k train split; 1,300 target-class images and the remaining 8,900 non-target images in the consolidated activation set for each binary RFM task |
| *RFM Training* | |
| Kernel | Laplace |
| Bandwidth | 200 |
| Regularization | $10^{-4}$ |
| Iterations | 5 |
| Top-$k$ eigenvectors | 50 saved; first signed eigenvector used for steering |

*Table 19.* **ImageNet effective guidance settings.**

| Target | $\lambda$ | $w_{\mathrm{RFM}}$ | $s$ | $\sigma_{\mathrm{end}}$ | $\sigma_t \in [\sigma_{\mathrm{R}}^{\mathrm{lo}}, \sigma_{\mathrm{R}}^{\mathrm{hi}}]$ | RFM steering layer | Top-$k$ | Metric |
| --- | --- | --- | --- | --- | --- | --- | --- | --- |
| 111, nematode | 2.0 | 0.5 | 12.0 | 3.0 | $[0, 30.0)$ | `input_blocks_15` | 1 | 83.2% acc., 138.5 FID |
| 222, kuvasz | 2.0 | 1.0 | 8.0 | 2.0 | $[0, 80.0)$ | `input_blocks_15` | 1 | 30.5% top-1, 74.2% top-5, 138.4 FID |
| 333, hamster | 1.0 | 0.5 | 4.0 | 10.0 | $[0, 80.0)$ | `input_blocks_15` | 1 | 91.8% acc., 56.3 FID |
| 444, tandem bicycle | 0 | 1.0 | 12.0 | – | $[0, 80.0)$ | `input_blocks_15` | 1 | 97.7% acc., 53.9 FID |

## H.7. CelebA-HQ Implementation

*Table 20.* **CelebA-HQ implementation details.**

| Parameter | Value |
| --- | --- |
| *Diffusion Model* | |
| Architecture | DDPM U-Net |
| Image resolution | 256×256 |
| Model checkpoint | `google/ddpm-ema-celebahq-256` |
| *Activation Collection* | |
| RFM steering layer | `mid_block` |
| Feature map resolution | 8×8 |
| Collection noise level | $\sigma \approx 1.0$ |
| Training data | CelebA-HQ 256×256 (30,000 images) |
| *Attribute-Specific Training Data* | |
| Gender | Female: 18k, Male: 10k |
| Age | Young: 20k, Old: 1.9k |
| Hair color | Black: 8k, Blond: 9k |
| *RFM Training* | |
| Kernel | Laplace |
| Bandwidth | 150 |
| Regularization | $10^{-3}$ |
| Iterations | 5 |
| Top-$k$ eigenvectors | 5 |

*Table 21.* **CelebA-HQ effective guidance settings.** Attribute order follows the row name; for example, in Female + Non-Blond, $\lambda_1, w_{\text{RFM}}^{(1)}$ are for Female and $\lambda_2, w_{\text{RFM}}^{(2)}$ are for Non-Blond. The $\sigma_{\text{end}}$ column is the noise-alignment cutoff; the $\sigma_t \in [\sigma_{\text{R}}^{\text{lo}}, \sigma_{\text{R}}^{\text{hi}}]$ column gives the RFM/CFG steering window.

| Combination | $\lambda_1$ | $w_{\text{RFM}}^{(1)}$ | $\lambda_2$ | $w_{\text{RFM}}^{(2)}$ | $s$ | $\sigma_{\text{end}}$ | $\sigma_t \in [\sigma_{\text{R}}^{\text{lo}}, \sigma_{\text{R}}^{\text{hi}}]$ | RFM steering layer | Top-$k$ | $N$ | Acc. | log-KID |
|---|---|---|---|---|---|---|---|---|---|---|---|---|
| *Gender + Hair Color* | | | | | | | | | | | | |
| Female + Non-Blond | +2.0 | -0.20 | -2.0 | -0.40 | 2.0 | 3.5 | [0, 3.5) | mid_block | 5 | 256 | 86.4 | -2.8688 |
| Female + Blond | +2.0 | -0.20 | +2.0 | +0.40 | 2.0 | 3.5 | [0, 3.5) | mid_block | 5 | 256 | 80.1 | -2.9751 |
| Male + Non-Blond | +2.0 | 0 | -2.0 | 0 | – | 3.5 | – | – | – | 256 | 98.4 | -1.8124 |
| Male + Blond | +1.5 | +0.40 | +1.5 | +0.40 | 2.0 | 3.5 | [0, 3.5) | mid_block | 5 | 256 | 68.4 | -1.7982 |
| *Gender + Age* | | | | | | | | | | | | |
| Young + Female | 0 | +0.24 | 0 | -0.15 | 2.0 | – | $[0.5, \infty)$ | mid_block | 5 | 256 | 100.0 | -3.1604 |
| Old + Female | +2.0 | +0.40 | +2.0 | -0.20 | 2.0 | 3.5 | [0, 3.5) | mid_block | 5 | 256 | 85.2 | -1.8920 |
| Young + Male | +2.0 | +0.32 | +2.0 | +0.40 | 2.0 | 3.5 | [0, 3.5) | mid_block | 5 | 256 | 98.8 | -1.9391 |
| Old + Male | 0 | +0.30 | 0 | +0.30 | 2.0 | – | $[0.5, \infty)$ | mid_block | 5 | 256 | 100.0 | -1.1736 |

## H.8. Fine-Grained Bird Species Implementation

*Table 22.* **Birds-525 fine-grained implementation details.**

| Parameter | Value |
|---|---|
| *Diffusion Model* | |
| Architecture | ADM U-Net (same as ImageNet) |
| Image resolution | 256×256 |
| *Activation Collection* | |
| RFM steering layer | input_blocks_15 |
| Feature map resolution | 8×8 |
| Collection noise level | $\sigma \approx 0.60$ |
| Training data | Birds-525 (Hugging Face) 160 images per class |
| *RFM Training* | |
| Kernel | Laplace |
| Bandwidth | 200 |
| Regularization | $10^{-4}$ |
| Iterations | 5 |
| Top-$k$ eigenvectors | 1 |
| *Target Species* | |
| Lucifer Hummingbird | Partial ImageNet overlap ("hummingbird" class) |
| Scarlet Macaw | Not in ImageNet |
| Fairy Tern | Not in ImageNet |
| Brown Headed Cowbird | Not in ImageNet |
| *Sampling* | |
| Sampler | DDIM |
| DDIM sampling steps | 100 |
| $\eta$ | 0.0 |
| RFM window $\sigma_t \in [\sigma_{\text{R}}^{\text{lo}}, \sigma_{\text{R}}^{\text{hi}}]$ | [0.01, 157.4] |
| Samples per evaluation | 256 |

*Table 23.* **Birds-525 effective guidance settings.**

| Species | $\lambda$ | $w_{\text{RFM}}$ | $s$ | $\sigma_{\text{end}}$ | $\sigma_t \in [\sigma_{\text{R}}^{\text{lo}}, \sigma_{\text{R}}^{\text{hi}}]$ | RFM steering layer | Top-$k$ | Metric |
|---|---|---|---|---|---|---|---|---|
| Lucifer Hummingbird | 3.0 | 0.7 | 5.0 | 10.0 | [0.01, 157.4] | input_blocks_15 | 1 | 21.5% acc., 24.76 FID |
| Scarlet Macaw | 3.0 | 0.3 | 5.0 | 10.0 | [0.01, 157.4] | input_blocks_15 | 1 | 28.1% acc., 104.1 FID |
| Fairy Tern | 2.0 | 0.3 | 3.0 | 10.0 | [0.01, 157.4] | input_blocks_15 | 1 | 2.7% acc., 93.48 FID |
| Brown Headed Cowbird | 1.0 | 0.5 | 5.0 | 2.0 | [0.01, 157.4] | input_blocks_15 | 1 | 3.9% acc., 65.72 FID |

## H.9. Stable Diffusion 1.5 Depth-of-Field Implementation

*Table 24.* **Stable Diffusion 1.5 depth-of-field implementation details.**

| Parameter | Value |
|---|---|
| *Diffusion Model* | |
| Model | `stable-diffusion-v1-5/stable-diffusion-v1-5` |
| Task | Shallow depth-of-field steering for text-conditioned generation |
| Sampler | DDIM |
| DDIM sampling steps | 50 |
| Text CFG | 7.5 |
| *Direction Data* | |
| Dataset | `svnfs/depth-of-field`, 1,200 images ($\sim$600 per class) |
| Labels | Binary shallow/deep DoF labels; target direction is shallow DoF |
| Evaluation prompts | First 25 COCO-Karpathy validation captions |
| Seeds per prompt | 42, 123, 256, 789 |
| *Activation Collection* | |
| RFM steering layer | `unet.down_blocks[2].resnets[-1]` |
| Direction extraction noise level | $\sigma \approx 0.344$ |
| Feature representation | Flattened activations, dimension 327,680 |
| *RFM Training* | |
| Kernel | Laplace |
| Bandwidth | 5000 |
| Iterations | 5 |
| *Guidance Settings* | |
| Noise-alignment coefficient $\lambda$ | 0 |
| RFM coefficient $w_{\text{RFM}}$ | 0.7 |
| RFM amplification scale $s$ | 4.0 |
| RFM window $\sigma_t \in [\sigma_{\text{R}}^{\text{lo}}, \sigma_{\text{R}}^{\text{hi}}]$ | $[0.041, 13.12]$ |
| *Evaluation* | |
| Metric | Depth Anything V2 foreground/background split + Laplacian variance |
| Reported aggregate | BG sharpness 1085.03 $\rightarrow$ 785.89; FG/BG 2.247 $\rightarrow$ 2.585 |

