# OpenReview forum: "General and Efficient Steering of Unconditional Diffusion Models"
_ICML.cc/2026/Conference — ICML 2026 regular_

### Official Review · Reviewer_p3rx · 2026-03-10

**Soundness:** 3
**Presentation:** 2
**Significance:** 3
**Originality:** 3
**Overall Recommendation:** 4
**Confidence:** 3

**Summary:**

This paper studies how to make the steering of unconditional diffusion models more efficient. The new method builds upon some key observations, like an identical steering direction can be applied across different samples and timesteps. This motivates the authors to divide the inference process into early- and late-timesteps and apply two different methods accordingly. The experimental results strongly support both the efficiency and effectiveness of the proposed method.

**Compliance With Llm Reviewing Policy:**

Affirmed.

**Final Justification:**

The authors have addressed my concerns. I feel my initial score can already reflect my positive opinion towards this submission, so I am not planning to change it.

**Key Questions For Authors:**

1. I acknowledge that the authors have evaluated the proposed method on a few different U-Net architecture configs. Could this evaluation be extended to DiT models? If it is impossible to do so during rebuttal, a brief discussion would also help.
2. Based on my understanding, I find a finding in a paper called "Not all diffusion model activations have been evaluated as discriminative features" to be potentially related to the model property observed in this work. Specifically, this paper finds that although diffusion backbones take a noisy variable as inputs and predict another noisy variable as outputs, they actually reconstruct an inner representation of the clean target image $x_0$. This might be able to explain why simple steering methods work in early timesteps. Of course, if this is in fact not related, I will not be unhappy if the authors honestly point it out.

**Limitations:**

yes

**Strengths And Weaknesses:**

Pros:
1. The proposed method is very efficient and effective, showing strong application potential.
2. The proposed method is based on novel empirical observations of model properties, giving it a solid base.

Cons: Generally, I feel the writing could use some refinement.
1. After reading, I find it a bit hard to recall some key takeaways. This might be because the overall organization is not clear enough. For example, when presenting the key model property observations, the authors do not number different observations to make them easy to distinguish. For another example, the two design choices appear not one-to-one for the two parts of the method (early- and late-stage guidance), which makes it a bit hard to follow. Ideally, I am expecting to see two properties, in correspondence to two design choices, which are in correspondence to early- and late-stage methods.
2. In the first paragraph of Introduction, I believe the authors are trying to rationale the study of unconditional diffusion steering by stating some drawbacks of conditional diffusion, like the inability to generalize to unseen conditions. However, this message is not conveyed clearly enough. This may make a reader question the necessity to study unconditional generation given the rapid advancement of condition diffusion.
3. In Abstract, the name Noise Alignment feels confusing and does not convey enough information. Furthermore, the detailed descriptions for Noise Alignment and Transferable Concept Vectors in Abstract feel a bit repeated.

---

> ### Author Rebuttal · Authors · 2026-03-31
>
> We thank the reviewer for their constructive suggestions. We address each point below.
>
> > *"Could this evaluation be extended to DiT models?"*
>
> **Extension to DiT models.** Indeed, our NA-RFM extends to DiT models. We apply our method to official pretrained **SiT-XL/2** (a transformer-based latent diffusion model), using null-class conditioning as the unconditional baseline. Below are preliminary results steering 3 middle blocks (out of 28 transformer blocks in total):
>
> | Class | NA-RFM acc | NA-RFM FID | Noise-align-only acc | Noise-align-only FID |
> |---|---|---|---|---|
> | 111 (nematode) | 52.7% | 202.4 | 46.9% | 197.0 |
> | 222 (kuvasz) | 28.5% | 172.3 | 2.7% | 197.3 |
> | 333 (hamster) | 81.6% | 129.1 | 1.6% | 227.7 |
> | 444 (tandem bicycle) | 82.4% | 103.5 | 0.4% | 261.8 |
> | Average | 61.3% | 151.8 | 12.9% | 220.9 |
>
> NA-RFM achieves **61.3% top-1 classification accuracy** for guided generation on SiT-XL/2, already outperforming TFG-4's 59.8% on U-Net-based diffusion (where NA-RFM achieves 75.8% on this U-Net-based model). We believe performance can be further improved with more thorough exploration of which blocks and layers to steer in the transformer architecture. Notably, noise alignment alone (PCA-based Gaussian guidance without RFM steering) already achieves nontrivial accuracy — 46.9% on nematode and 12.9% average — demonstrating that our PCA-based guidance transfers to latent-space DiT models. Adding RFM activation steering then boosts accuracy dramatically on the remaining classes, while also improving FID (151.8 vs 220.9 average). We include a visualization of generated images in the following anonymous link: ​​https://anonymous.4open.science/r/27977/dit.png. We will be happy to include such experiments in the revision.
>
>
>
> > *"I find a finding in 'Not all diffusion model activations have been evaluated as discriminative features' to be potentially related."*
>
> **Connection to Meng et al. (NeurIPS 2024).** Thank you for this reference. Meng et al. study the quality of different internal activations within diffusion U-Nets as discriminative features, focusing on dense pixel-level tasks (segmentation, correspondence) at a fixed timestep. Their key finding is that activation quality varies across U-Net blocks, with early up-stage blcok activations appearing qualitatively cleaner. This internal denoising capability could be related to why discriminative directions exist in the activation space even at noisy timesteps. In our response to Reviewer aDr1, we include a conceptual understanding for this point.
> > *"I find it a bit hard to recall some key takeaways...the overall organization is not clear enough."*
> > *"In the first paragraph of Introduction...this message is not conveyed clearly enough."*
> > *"In Abstract, the name Noise Alignment feels confusing...descriptions feel a bit repeated."*
>
> **Regarding writing refinement.** Thank you for the suggestions. We will better organize the supporting evidence and improve the presentation following the suggestions on the abstract and introduction.
>
>  Meng, Benyuan, et al. "Not all diffusion model activations have been evaluated as discriminative features." NeurIPS (2024)

---

> > ### Author Rebuttal · Reviewer_p3rx · 2026-04-03
> >
> > The authors have addressed my concerns. I feel my initial score can already reflect my positive feeling towards this submission, so I am not planning to change it.

---

### Official Review · Reviewer_CzaD · 2026-03-12

**Soundness:** 3
**Presentation:** 2
**Significance:** 3
**Originality:** 3
**Overall Recommendation:** 5
**Confidence:** 4

**Summary:**

This paper aims to efficiently guide unconditional diffusion models at inference time without computing gradients and backpropagating. The authors identify that (1) highly corrupted images early on still encode coarse semantics that could be used for signal guidance for class information, (2) directions from later timesteps (better recovered images) remain effective with intermediate ones, allowing for fine-grained semantic steering. From these observations, this paper proposes a light-weight two-stage steering method during inference, which only requires fast vector arithmetics and shows significant speedup & improved quality across various experiments.

**Compliance With Llm Reviewing Policy:**

Affirmed.

**Final Justification:**

I am keeping my original "Accept" score.

**Key Questions For Authors:**

Please see the above Strengths and Weaknesses Section.

**Limitations:**

Yes in the last section; please also consider adding the comments that cannot be immediately resolved in this paper to the limitations.

**Strengths And Weaknesses:**

**Strengths**:

* The method is well motivated, and the flow of this paper is smooth, making it easy to follow. The authors present two key observations behind their proposal before introducing the technical details, which establishes the intuition smoothly.

* The technical details are explained clearly in two stages, effectively motivated by their empirical findings. The computational complexity for offline preparation is carefully analyzed to support the claimed efficiency of the method during inference.

* Empirical results sufficiently validate the effectiveness of the proposed method in many scenarios, including larger scale, multi-attribute guidance, and OOD images not seen during training. Various ablations are performed to further the understanding.

I do not spot any major weaknesses that undermine the novelty of this paper, but I have some following minor suggestions and questions:

* The abstract mostly covers the two observations driving this new approach and then mentions experiments. It would be better to include a high-level description of your method here.
* The introduction makes readers assume that the method only operates during inference, but it turns out that there are two sources of offline compute from calculating class-conditional PCA statistics and collecting forward-pass activations. Please consider making it clear early in the introduction to avoid confusion.
* I believe that there is a missing cluster of related works on steering vectors in language models. The authors briefly mention this line of works to motivate their method, but I find it necessary to include a new subsection discussing the difference in steering language vs vision models and potential generalization of this idea to LLMs.
* If I am understanding correctly, the $\hat{v}_c$ in Equation 9 is an estimated value of $v_c$ in Line 271? In Algorithm 1, notations in noise alignment window and RFM window do not seem to show up earlier. Let's fix the technical details?
* Consider giving this method a name? (It is named NA-RFM in the Experiments section, but nothing in the introduction or abstract).

Overall I believe this work would be a valuable addition to the venue. I will give it a 3 in Soundness, Significance, Originality, and a 2 in Presentation for the unclear notations.

---

> ### Author Rebuttal · Authors · 2026-03-31
>
> We thank the reviewer for their thoughtful feedback. We appreciate the suggestions on presentations and will incorporate them in the revision. Below, we address the other points raised.
>
> > *"The variable in Equation 9 is an estimated value of the one in Line 271? In Algorithm 1, notations...do not seem to show up earlier."*
>
> Yes, $\hat{v}_c$ in Eq. 9 is the estimated steering direction defined at Line 271. Thank you for pointing out this, we will make this explicit. We will also define the noise alignment and RFM active windows in the method introduction before they appear in Algorithm 1.
>
>
> > *"I believe that there is a missing cluster of related works on steering vectors in language models. The authors briefly mention this line of works to motivate their method, but I find it necessary to include a new subsection discussing the difference in steering language vs vision models and potential generalization of this idea to LLMs."*
>
> Thank you for the suggestions. We will extend the related work discussion on language model steering. We will also add a new delicate subsection discussing the differences between LLM steering and our diffusion model approach in the revision.

---

> > ### Author Rebuttal · Reviewer_CzaD · 2026-04-03
> >
> > Thank you for the responses! I will keeping my original "Accept" score.

---

### Official Review · Reviewer_adXf · 2026-03-13

**Soundness:** 3
**Presentation:** 3
**Significance:** 3
**Originality:** 3
**Overall Recommendation:** 5
**Confidence:** 3

**Summary:**

In this work, authors present NA-RFM, a framework to steer unconditional diffusion models without gradient computation at test time. Authors utilize RFMs to find discriminative editing directions and use the observation that class information emerges early for guided generation. Authors demonstrate the effectiveness of NA-RFM on various datasets/benchmarks (CIFAR-10, ImageNet, CelebA-HQ, Birds-525). Experiments demonstrate that NA-RFM performs better than the competing methods while simultaneously being more efficient in terms of sampling speed.

**Compliance With Llm Reviewing Policy:**

Affirmed.

**Final Justification:**

In the rebuttal period, the authors clarified the motivation of the work and addressed my concern regarding the need for steering unconditional diffusion models. In particular, they provided evidence that the proposed method does not rely on unconditional models and can also steer conditional diffusion models, as demonstrated through the depth-of-field experiments. The revised framing, which focuses on “steering toward conditions that are unseen at training time,” helps better contextualize the contribution and aligns with the presented results.

With the understanding that this clarification will be clearly reflected in the introduction and abstract of the revised manuscript, I find the response satisfactory. As such, I update my assessment and increase my score to accept (5).

**Key Questions For Authors:**

* Related to the weakness point above, could the authors comment on realistic settings where one needs to steer unconditional diffusion models?
* Some suggestions for contextualizing the work better.  Early emergence of semantic layout/class information and their different diffusion stages is explored in various works (see [1-5]). I would recommend including them in related works.
* Could the authors comment on intuitively why there is a significant gap between "Middle" U-Net block vs. "Encoder" and "Decoder" blocks (in Figure 6, Appendix B)?
* line 605: incorrect reference (??).

***
***References:***

[1] Li, Shuangqi, et al. "All Seeds Are Not Equal: Enhancing Compositional Text-to-Image Generation with Reliable Random Seeds." arXiv preprint arXiv:2411.18810 (2024).

[2] Hertz, Amir, et al. "Prompt-to-prompt image editing with cross attention control." arXiv preprint arXiv:2208.01626 (2022).

[3] Patashnik, Or, et al. "Localizing object-level shape variations with text-to-image diffusion models." Proceedings of the IEEE/CVF international conference on computer vision. 2023.

[4] Tinaz, Berk, Zalan Fabian, and Mahdi Soltanolkotabi. "Emergence and Evolution of Interpretable Concepts in Diffusion Models." The Thirty-ninth Annual Conference on Neural Information Processing Systems.

[5] Mahajan, Shweta, et al. "Prompting hard or hardly prompting: Prompt inversion for text-to-image diffusion models." Proceedings of the IEEE/CVF Conference on Computer Vision and Pattern Recognition. 2024.

**Limitations:**

yes

**Strengths And Weaknesses:**

***Strengths:***
* The paper is writted very well. It is easy to follow and understand the methodology.
* The reported numbers are very impressive. Significant performance boost compared to baselines while keeping it cheap in terms of inference time needed.
* Experiments and ablations studies are comprehensive.

***
***Weaknesses:***
* On a broader scope, I am not sure how important it is to steer unconditional diffusion models. At least in the context of image generation/editing, there are various open sourced and state-of-the-art generative models that are already trained to accept conditioning inputs.
* The work could be contextualized better in terms of some of its claims (i.e. early emergence of class information).
* Please see the questions below.

---

> ### Author Rebuttal · Authors · 2026-03-31
>
> We appreciate the reviewer's constructive feedback. We address the raised points below.
>
> > *"I am not sure how important it is to steer unconditional diffusion models."*
> > *"Could the authors comment on realistic settings where one needs to steer unconditional diffusion models?"*
>
>
> We would like the clarify that our method does **not** require an unconditional model. Our method can be straightforwardly applied to a conditional diffusion model to incorporate more conditions even when those are not seen at training time. We should have made this more clear in the paper – our method can be used to steer (a conditioned or unconditioned) model towards unseen conditions. We demonstrate this concretely with the experiment below.
>
>
>
> **Experiment description:** We apply our method to a text conditioned diffusion model (Stable Diffusion 1.5) to steer the depth-of-field (DoF) attribute. We demonstrate that NA-RFM can steer toward shallower DoF on Stable Diffusion 1.5 for generations conditioned on text prompts.
>
> *1. Direction learning.* We use the svnfs/depth-of-field dataset from HuggingFace, which provides binary shallow/deep DoF labels across diverse real-world objects (~600 images per class). We extract activations at the last encoder block before the bottleneck and learn an RFM direction.
>
> *2. Evaluation metric.* We estimate per-pixel depth using Depth Anything V2 [1] and segment each generated image into foreground (top 40% depth) and background (bottom 60%). We compute Laplacian variance on each region as a sharpness proxy, and report the foreground-to-background (FG/BG) sharpness ratio — higher values indicate sharper foreground relative to a blurred background, mimicking shallow DoF.
>
> *3. Steering results.* Across the first 25 prompts from COCO dataset (4 generations for each prompt, 100 images in total), steering achieves on average:
> - Background sharpness reduced by 27.6%
> - FG/BG sharpness ratio increases from 2.25 to 2.58
> - The transition is continuous — object identity and composition are roughly preserved as DoF varies, see qualitative visualizations in link (https://anonymous.4open.science/r/27977/dof.png).
>
> This demonstrates that NA-RFM works with conditional models to steer implicit features learned from examples. We also want to emphasize that our model does not require an explicit “concept”, and all we need are just some examples with labels (e.g, positive or negative), which can be useful in practice, as sometimes it can be hard to articulate a concept.
>
>
>
> > *"Could the authors comment on intuitively why there is a significant gap between Middle U-Net block vs Encoder and Decoder blocks?"*
>
> The middle block operates at the spatial bottleneck (e.g., 4x4 for 32x32 inputs), where spatial information is maximally compressed. Two possible reasons for the gap: (1) limited spatial resolution makes steering less expressive; (2) skip connections bypass the bottleneck — decoder blocks combine encoder features directly, dampening the middle block's downstream influence. Encoder and decoder blocks retain richer spatial structure, explaining the consistent gap.
>
>
> > *"The work could be contextualized better in terms of some of its claims (i.e. early emergence of class information)."*
>
> Thank you for the relevant references. We will add them to the related work section in our revision.
>
> > *"line 605: incorrect reference (??)"*
>
> Thank you for pointing out. The intended figure (anonymous link: https://anonymous.4open.science/r/27977/abl.png) was inadvertently omitted. We will restore it in the revision.
>
> [1] Yang, Lihe, et al. "Depth anything v2." NeurIPS (2024):

---

> > ### Author Rebuttal · Reviewer_adXf · 2026-04-03
> >
> > I would like to thank the authors for their time and effort in responding to my concerns and questions, and for running new experiments on steering conditional diffusion models.
> >
> > My concern about motivating the need for unconditional diffusion models is not fully resolved. I concur with the reviewers that the proposed method can work with conditional diffusion models as well, which is a nice bonus. This is demonstrated well through the depth-of-field experiments. However, the whole paper, including the title, is written from the perspective of controlling unconditional models. Therefore, without rewriting the full motivation of the paper, I believe the authors should do a better job of motivating unconditional models, as also raised by Reviewer p3rx. Can the authors comment on this?

---

> > > ### Author Response · Authors · 2026-04-04
> > >
> > > We thank the reviewer for the follow-up question. The main motivation of our work is to enable efficient steering toward conditions that are **unseen** at training time. We use unconditional diffusion models as a clean testbed, since in that setting, all conditions are effectively unseen. We agree, however, that the opening of the introduction puts too much emphasis on “unconditional diffusion,” which can obscure the broader contribution. In the revision, we will revise the introduction to make it clearer from the outset that the broader goal is to steer a pretrained diffusion model toward unseen conditions after training, and that guidance for unconditional models is a clean special case for evaluation; We will also adjust the abstract to reflect this framing more clearly.

---

### Official Review · Reviewer_aDr1 · 2026-03-13

**Soundness:** 2
**Presentation:** 3
**Significance:** 3
**Originality:** 3
**Overall Recommendation:** 5
**Confidence:** 4

**Summary:**

A broad theme addressed by this article is how to add effective controllability to unconditional diffusion models without retraining them as conditional models and without paying the high inference-time cost of per-step gradient-based guidance. Overall, the main problem investigated by this article is whether one can steer unconditional diffusion efficiently using only offline-computed signals and forward-pass-time interventions. The paper proposes NA-RFM, a two-stage steering framework: noise alignment at high noise levels and RFM-based activation steering at later stages. It is evaluated on CIFAR-10, ImageNet, CelebA, and a fine-grained bird-species task, where it achieves stronger guidance accuracy, better FID/KID, and faster inference than TFG and other training-free gradient-based baselines.

**Compliance With Llm Reviewing Policy:**

Affirmed.

**Final Justification:**

The author supplemented with additional experimental results, evaluating the experiment under the same knowledge-guided setting. The results showed that even when the baseline method was augmented with teacher guidance, D2-LOS remained superior. This indicates that the value lies not in the presence or absence of a teacher model, but in how to dynamically utilize the knowledge of the teacher model under a long-tail distribution, fully addressing my concerns.Therefore, I will raise my score to Accpet.

**Key Questions For Authors:**

1.	Since the main baselines are gradient-based guidance methods, can the authors clarify how their method compares to a broader set of non-gradient or activation-space steering approaches for diffusion?
2.	The paper argues that clean-time forward activations produce transferable directions. Can the authors provide a stronger conceptual explanation for why these directions remain effective across later timesteps of reverse sampling?

**Limitations:**

The main limitations are that the method depends on labeled data or class statistics for direction/statistics construction, its strongest evidence is still within image-class guidance settings, and the broader generality of the approach beyond the tested model/dataset families remains to be established.

**Strengths And Weaknesses:**

Strengths
1. The paper studies an important and practically relevant problem.
2. The method is conceptually clear and easy to follow.The two-stage design matches the empirical probing results about where class information is available and where activation-level steering is reliable.
3. The probing experiments are a strong part of the paper.
4. The empirical gains are strong, with notably better guidance accuracy, FID, and inference speed than TFG-style gradient-based guidance baselines across CIFAR-10 and ImageNet.
5. The paper includes useful ablations, including training noise level, block selection, and guidance-window analyses, which help support the practical design choices.

Weaknesses
1. The evaluation focuses heavily on guidance accuracy and FID/KID relative to gradient-based guidance baselines, but there is less analysis of possible trade-offs such as diversity loss, calibration of the evaluation classifiers, or sensitivity to dataset-specific hyperparameters.
2. The theory is relatively light. The paper is well motivated empirically, but it does not provide a strong formal account of why the learned RFM directions should transfer across timesteps or why the combined two-stage scheme should preserve generation quality while improving control.

---

> ### Author Rebuttal · Authors · 2026-03-31
>
> We thank the reviewer for their insightful feedback. Below, we will address the major points raised.
>
> **Regarding a stronger conceptual explanation.**  Regarding the reviewer's question on a "stronger conceptual explanation" for why RFM directions transfer across timesteps, indeed, we can provide a partial theoretical justification by analyzing a special case (described below). Concretely, under binary isotropic Gaussian mixture data, we show the RFM learned direction is constant across all timesteps. With a Gaussian proxy model for the activations of general data, we show the RFM learned direction in activation space remains robust across intermediate noise levels. We will include this in the revised manuscript.
>
> Suppose the data follow a binary Gaussian mixture with shared covariance: $x_0 \mid y{=}k \sim \mathcal{N}(\boldsymbol{\mu}_k, \Sigma)$, $\Delta\boldsymbol{\mu} := \boldsymbol{\mu}_1 - \boldsymbol{\mu}_2$. The Bayes log-odds $\ell(x) = \log p(y{=}1 \mid x) - \log p(y{=}2 \mid x)$ is affine in $x$: because the two classes share $\Sigma$, the quadratic terms $x^T \Sigma^{-1} x$ cancel, leaving $\ell(x) = x^T \Sigma^{-1}\Delta\boldsymbol{\mu} + c$. The gradient $\nabla_x \ell = \Sigma^{-1}\Delta\boldsymbol{\mu}$ is therefore constant, so the AGOP (average gradient outer product) is rank 1 with top eigenvector proportional to $\Sigma^{-1}\Delta\boldsymbol{\mu}$. RFM is a backpropagation-free method that aligns with this AGOP computation.
>
> Under the forward process $x_t = \alpha_t x_0 + \beta_t \epsilon$, the noisy observations remain a binary GMM: $x_t \mid y{=}k \sim \mathcal{N}(\alpha_t \boldsymbol{\mu}_k,\, \alpha_t^2 \Sigma + \beta_t^2 I)$. The Bayes classifier remains linear, and the AGOP direction becomes $\mathbf{v}_t \propto (\Sigma + \sigma_t^2 I)^{-1}\Delta\boldsymbol{\mu}$ where $\sigma_t^2 = \beta_t^2/\alpha_t^2$. When $\Sigma$ is isotropic ($\Sigma = \gamma I$), $\mathbf{v}_t \propto \Delta\boldsymbol{\mu}$ for all $t$, that is, the direction is exactly preserved across timesteps. In the anisotropic case, the direction varies continuously with $\sigma_t^2$: drift is small when $\sigma_t^2$ is moderate relative to the eigenvalues of $\Sigma$ along which $\Delta\boldsymbol{\mu}$ has nontrivial projection.
>
> If we model activations at a semantic layer as a class-conditional Gaussian mixture with noise-dependent covariance $\mathbf{h} \mid y{=}k, t \sim \mathcal{N}(\boldsymbol{\mu}_k^h,\, \Sigma^h + \eta_t^2 I)$. Then the same analysis above carries over, with the effective activation-space noise $\eta_t$ in place of input-space $\sigma_t$. As Reviewer p3rx notes empirically in [M2024] semantic layers of diffusion U-Nets reconstruct approximately clean features even from noisy inputs, suggesting $\eta_t$ may be smaller than $\sigma_t$, which would make the AGOP direction even more stable in activation space than data space.
>
> **Regarding diversity and evaluation.** As reported in the paper, in addition to the best accuracy, our NA-RFM also achieves the best FID scores across tasks which indicates good diversity. However, we have also run additional new experiments to measure generative recall defined in [K2019] as another way for measuring diversity:
>
> | Method | Accuracy | FID ↓ | Recall ↑ |
> |---|---|---|---|
> | NA-RFM | 96.6% | 41.4 | 0.442 |
> | Classifier Guidance | 86.0% | 47.0 | 0.430 |
> | TFG-4 | 77.1% | 73.9 | 0.362 |
>
> As we see, our method also achieves strong recall while achieving the best accuracy and FID simultaneously.
>
>
> **Regarding calibration of the evaluation classifiers:** We adopt the same evaluation protocol as baseline TFG with separated labeling and evaluation networks. To address your comments, we have now also tested two additional classifiers for CIFAR-10 and ImageNet dataset, and the performance remains similar. In particular, for CIFAR-10, we also tested on ResNet56 and VGG19-BN with 96.5% and 97.2% accuracy, respectively, aligning with our reported 96.6% using ConvNeXt-Tiny ; for ImageNet, we tested ResNet-50 and ConvNeXt-Base with 71.8% and 75.5%, aligning with our reported 75.8% using DeiT-Small.
>
>
> **Clarify how our method compares to a broader set of non-gradient or activation-space steering approaches for diffusion.** NA-RFM learns directions from labeled data via the forward process and offline RFM — no reverse sampling, DDIM inversion, or text conditioning. This distinguishes it from h-space editing which uses DDIM inversion, cross-attention methods which require text conditioning, and approaches that backpropagate through the model.
>
>
> [K2019] Kynkäänniemi et al., "Improved precision and recall metric for assessing generative models," NeurIPS 2019.
>
> [M2024] Meng, Benyuan, et al. "Not all diffusion model activations have been evaluated as discriminative features." NeurIPS 2024.

---

### Decision · Program_Chairs · 2026-04-30

**Decision:**

Accept (regular)

**Comment:**

This paper introduces NA-RFM, a two-stage, gradient-free framework for steering unconditional diffusion models by combining early-stage noise alignment with activation-space steering using Recursive Feature Machines (RFMs). The reviewers appreciated the method's conceptual clarity, inference efficiency, and strong empirical performance, noting significant improvements in both guidance accuracy and FID over gradient-based baselines. During the rebuttal, the authors addressed primary concerns regarding the motivation for steering unconditional models by demonstrating that NA-RFM can also steer conditional models (e.g., Stable Diffusion) toward attributes unseen during training, such as depth-of-field. They further strengthened the work by providing theoretical justification using Gaussian mixture models and demonstrating generalization to transformer-based (DiT) architectures. While some reviewers noted minor presentation issues, they agreed that the added experiments and clarifications resolved their major concerns. Given the unanimous positive ratings and the efficient inference-time guidance mechanism, the submission is recommended for acceptance.